# Joint Learning of 2D-3D
# Weakly Supervised Semantic Segmentation

**Hyeokjun Kweon**
KAIST
0327june@kaist.ac.kr

**Kuk-Jin Yoon**
KAIST
kjyoon@kaist.ac.kr

## Abstract

The aim of weakly supervised semantic segmentation (WSSS) is to learn semantic segmentation without using dense annotations. WSSS has been intensively studied for 2D images and 3D point clouds. However, the existing WSSS studies have focused on a single domain, *i.e.* 2D or 3D, even when multi-domain data is available. In this paper, we propose a novel joint 2D-3D WSSS framework taking advantage of WSSS in different domains, using classification labels only. Via projection, we leverage the 2D class activation map as self-supervision to enhance the 3D semantic perception. Conversely, we exploit the similarity matrix of point cloud features for training the image classifier to achieve more precise 2D segmentation. In both directions, we devise a confidence-based scoring method to reduce the effect of inaccurate self-supervision. With extensive quantitative and qualitative experiments, we verify that the proposed joint WSSS framework effectively transfers the benefit of each domain to the other domain, and the resulting semantic segmentation performance is remarkably improved in both 2D and 3D domains. On the ScanNetV2 benchmark, our framework significantly outperforms the prior WSSS approaches, suggesting a new research direction for WSSS.

## 1 Introduction

Semantic segmentation is a fundamental problem in the field of computer vision. Recently, learning-based semantic segmentation methods have achieved remarkable performance. However, they usually rely on strong supervision such as pixel-wise (2D) or point-wise (3D) dense semantic labels. Since dense labels are time-consuming and labor-intensive to annotate, the labeling process has remained the main obstacle to applying semantic segmentation to real problems. Weakly supervised semantic segmentation (WSSS) has flourished to learn semantic segmentation by using inexpensive weak labels (*e.g.* bounding boxes or class labels).

WSSS using class labels has been widely studied in the 2D domain (image) and has recently also been scaled into the 3D domain (point cloud). For acquiring pixel- or point-wise class predictions without any spatial supervision, the existing 2D and 3D WSSS studies have employed Class Activation Maps (CAM) [35] extracted from a classifier. Since CAM usually localizes the discriminative regions of each class (which be used as cues for classification by classifier), one can acquire pseudo-labels for semantic segmentation from the CAM. While the studies have shown promising results, they focus on using the data of a single domain and place less emphasis on the information of other domains that might have complementary properties.

Fig. 1 visualizes the complementary advantages of 2D and 3D WSSS. With rich and dense features from a CNN-based architecture, 2D WSSS succeeds in the semantic perception of images. However, the 2D CAM is usually not sharp at the object boundary and the segmentation result tends to be imprecise as shown in (b3). On the other hand, 3D WSSS is advantageous in terms of segmentation, since the input point cloud contains the 3D geometry of the scene. However, due to the severe class

36th Conference on Neural Information Processing Systems (NeurIPS 2022).

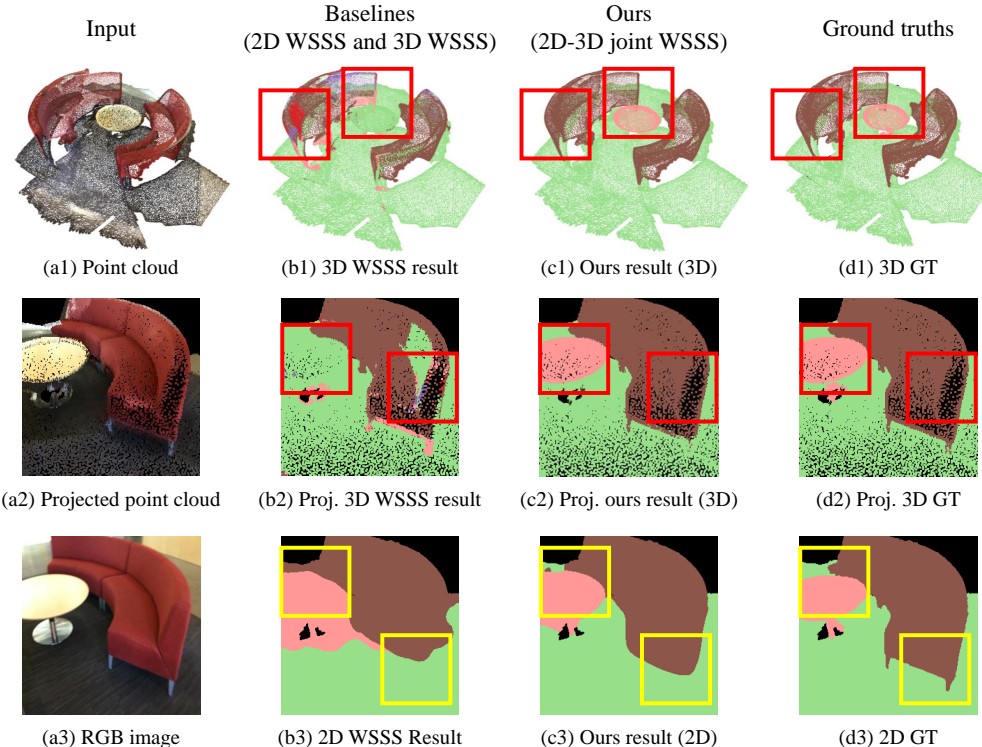

| Input | Baselines (2D WSSS and 3D WSSS) | Ours (2D-3D joint WSSS) | Ground truths |
|---|---|---|---|
| (a1) Point cloud | (b1) 3D WSSS result | (c1) Ours result (3D) | (d1) 3D GT |
| (a2) Projected point cloud | (b2) Proj. 3D WSSS result | (c2) Proj. ours result (3D) | (d2) Proj. 3D GT |
| (a3) RGB image | (b3) 2D WSSS Result | (c3) Ours result (2D) | (d3) 2D GT |

Figure 1: Comparison of results from single 2D or 3D domain WSSS and the proposed multi-domain WSSS. The second column shows the results of 3D WSSS (**b1**, **b2**) and 2D WSSS (**b3**). 3D WSSS *misclassifies* the semantics of the objects (red boxes), while the 2D WSSS generates *imprecise* prediction at object boundaries (yellow boxes). Compared to the uni-domain results, **the proposed 2D-3D joint WSSS framework** achieves much more precise and complete semantic segmentation results in both domains (**c1-c3**), by unifying the benefits of multi-domain data.

imbalance in 3D data (e.g. *wall* or *floor* exist in almost every scene) and the sparse nature of the point cloud, 3D WSSS difficulty in predicting the semantics as in (b1) and (b2). The results of 2D and 3D WSSS show different tendencies originating from the distinct domains, and they are complementary for achieving semantic segmentation of the scene.

Upon this background, we propose a novel 2D-3D WSSS framework that jointly targets both domains of image and point cloud. For unifying the complementary strengths of both domains, we propagate the benefit of one domain to the other domain via self-supervision, and vice versa. It enables our framework to **individually** perform 2D and 3D WSSS, without requiring the paired input data in the inference phase. We obtain a 2D pseudo-label from the 2D CAM of the image classifier and make 3D CAM follow it, which enhances the semantic perception of the point cloud classifier. On the other hand, to transfer the segmentation capability of 3D CAM to 2D, we construct a correlation matrix between the 3D features. The image classifier is then trained to generate 2D features having a similar correlation matrix to that of the 3D, which relieves the impreciseness problem of 2D CAM. In both directions, to prevent the networks from being spoiled by inaccurate self-supervision, the framework obtains a confidence score for each region and rejects erroneous predictions.

With extensive qualitative and quantitative experiments, we verify that the proposed joint framework effectively transfers the benefit of each domain to the other domain as visualized in (c1-c3) of Fig. 1. As a result, in both image and point cloud domains, the semantic segmentation performance is remarkably enhanced simultaneously. For example, on the ScanNetV2 [5] benchmark, our method significantly outperforms the existing WSSS studies, demonstrating its superiority. Further, with an experiment on generalization capability to the S3DIS dataset, we verify that the gain of our joint framework could be extended to the unobserved data. The proposed framework is the first multi-domain approach for the field of WSSS, which is not only effective for learning both semantics and segmentation but also practical for being applied in real cases. We strongly believe that 2D-3D joint learning is a promising direction for conquering semantic segmentation with class labels only.

Table 1: Summary of related works in the field of WSSS. Compared to the prior single 2D or 3D domain works, our method targets both 2D and 3D domains by using image-level and scene-level class labels only. CLS and SS denote class labels and semantic segmentation labels, respectively.

| Domains | 2D | 3D | | | 2D+3D |
|---------|-----|------|-----------|-----------|--------|
| Methods | [11, 13, 1, 32, 24, 16, 3, 7, 25] | [20, 31, 8, 33, 34] | MPRM[29] | WyPR[22] | Proposed |
| Weak Labels | CLS (image) | SS (few points) | CLS (subcloud) | CLS (scene) | CLS (image and scene) |

## 2 Related works

In this section, we introduce the existing studies on 2D/3D WSSS and clarify the uniqueness of the proposed WSSS method that targets both domains in a simultaneous manner. Table 1 shows the setting of the weak labels we used and those of existing studies. Note that the proposed framework is the first method that exploits the complementary strengths of WSSS in the 2D and 3D domains, using image- and scene-level class labels only.

**WSSS in 2D domain**     2D WSSS with image-level class labels only [11, 13, 1, 32, 24, 16, 3, 7, 25] has been widely studied for the last decade. Since the class labels do not provide any spatial supervision, most WSSS studies have exploited a CAM for localizing the regions of each class. Although the CAM can serve as an acceptable estimate for localization, they are usually activated in the most discriminative regions only, rather than capturing whole object regions. Moreover, the CAM is sensitive to the scale variance of the input image and less precise at the boundary of the objects. To relieve these issues, existing works have proposed various techniques such as refining [11, 13, 1], adversarial erasing [32, 24, 16], sub-categorical classification [3], and cross-image attention [7, 25]. However, their problem definition and benchmark dataset [6, 18] mainly focus on the 2D domain and less consider 3D structures of the scene, which are valuable for solving the aforementioned issues.

**WSSS in 3D domain**     Point-wise dense semantic segmentation labeling for a 3D point cloud is notorious for being difficult to annotate (22.3 min) [29]. To resolve this, several studies [20, 31, 8, 33, 34] have been conducted to learn semantic segmentation with only partially labeled points. However, since acquiring partial point-wise annotation is still expensive even under the "One Thing One Click" setting (2 min) [20], a few studies have been conducted for exploiting less expensive class labels (15 sec) [29]. MPRM [29] samples subclouds from the full scene and annotates them, since using subcloud-level class labels can relieve the class imbalance issue (*e.g. wall* or *floor*) commonly appearing in almost every scene). Recently, WyPR [22] jointly targets semantic segmentation and object detection of a point cloud with scene-level class labels only. The studies have shown the feasibility of WSSS in the 3D domain; however, due to the sparse and irregular nature of the point cloud, the performance of 3D WSSS is limited compared to the 2D WSSS.

**Semantic segmentation in combined 2D-3D domains**     To exploit both 2D and 3D data for semantic segmentation, several semantic segmentation studies have been conducted for uni-directional[12, 4] or bi-directional[10] feature projection between the 2D and 3D domains, in a *fully-supervised setting*. [28, 15] leveraged the 2D semantic segmentation for learning the 3D semantic segmentation with multi-view or virtual-view settings; however, they still require a dense 2D GT. [26] proposed to use 3D information as guidance for enhancing 2D WSSS using bounding boxes. Recently, [9, 17] incorporated 3D priors for self-supervised learning of 2D networks to achieve 3D-aware scene recognition. Unlike the existing studies, our framework is the first WSSS approach that jointly unifies the complementary benefits of 2D and 3D domains, by using the image-level and scene-level class labels only.

## 3 Method

In this section, we introduce the proposed 2D-3D joint framework for learning WSSS in both domains. Figure 2 shows our overall framework.

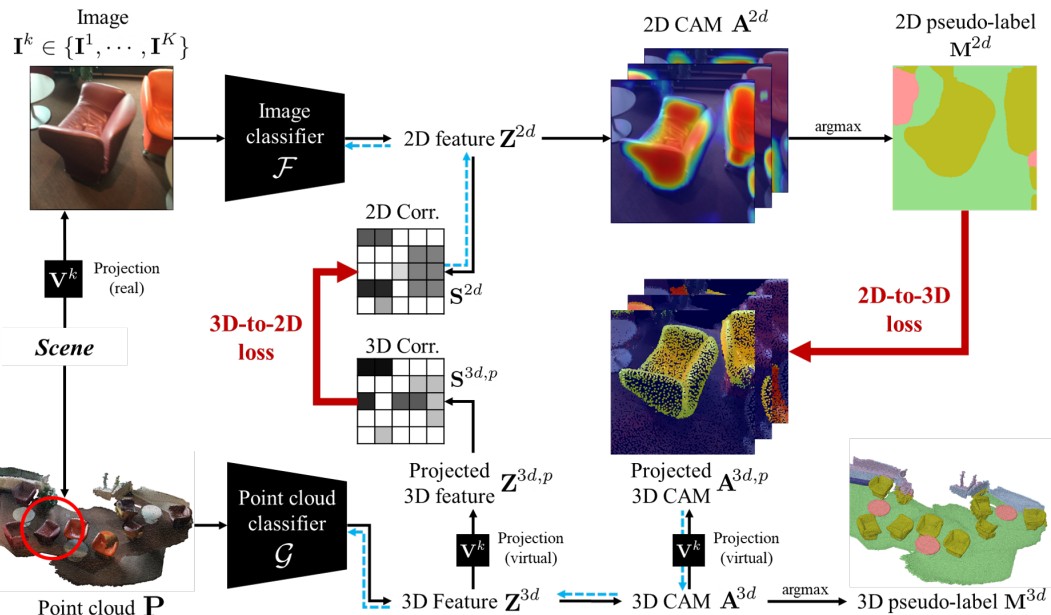

Figure 2: Our overall framework. As inputs, we use a point cloud $\mathbf{P}$ and an image $\mathbf{I}^k$ sampled from a set of images corresponding to the point cloud, where the projection for the image is given as $\mathbf{V}^k$. Note that the image is captured from the real scene. An image classifier $\mathcal{F}$ and a point cloud classifier $\mathcal{G}$ are trained to extract 2D outputs ($\mathbf{Z}^{2d}$ and $\mathbf{A}^{2d}$) and 3D outputs ($\mathbf{Z}^{3d}$ and $\mathbf{A}^{3d}$), respectively. To bridge the 2D and 3D domains, the 3D outputs are projected onto the image plane by using $\mathbf{V}^k$. For **3D-to-2D**, we construct a correlation matrix of 2D features ($\mathbf{S}^{2d}$) and make it to follow that of the projected 3D features ($\mathbf{S}^{3d,p}$). For **2D-to-3D**, we generate 2D pseudo-label $\mathbf{M}^{2d}$ from 2D CAM and make projected 3D CAM follow it. Note that blue dashed lines denote back-propagation of the losses.

## 3.1 Acquiring CAM of each dimension

Following the prior works with the class labels only, we exploit a CAM to localize the class regions from the image or point cloud. We assume that a set of $K$ images $\mathbf{I} = \{\mathbf{I}^1, \cdots, \mathbf{I}^K\}$ and a point cloud $\mathbf{P}$ are obtained from a certain scene. $\mathbf{I}^k$ is an RGB image with a size of $H \times W$, and $\mathbf{P}$ is a set of $N$ points where each point is represented by its 3D coordinate (XYZ) and color (RGB), as in [22]. A view matrix $\mathbf{V}^k$ (perspective projection matrix) of the camera for the image $\mathbf{I}^k$ is also given, as in many point cloud datasets [5, 2]. As weak supervision, we use the image-level class label $\mathbf{t}^{2d}$ and scene-level class label $\mathbf{t}^{3d}$, where both labels are represented as $C$-dimensional binary vectors. $C$ denotes the total number of classes, and the classes are non-exclusive since multiple classes can co-exist in a single image (or point cloud).

As visualized in Fig. 2, we train an image classifier $\mathcal{F}$ to extract a 2D feature map $\mathbf{Z}^{2d} \in \mathbb{R}^{D \times H \times W}$. A single linear layer having $D$ input channels and $C$ output channels then serves as a classification head for making pixel-wise class prediction $\mathbf{A}^{2d} \in \mathbb{R}^{C \times H \times W}$. Similarly, a point cloud classifier $\mathcal{G}$ is trained to provide point-wise features $\mathbf{Z}^{3d} \in \mathbb{R}^{D \times N}$ and point-wise class prediction $\mathbf{A}^{3d} \in \mathbb{R}^{C \times N}$. Here, we directly employ $\mathbf{A}^{2d}$ and $\mathbf{A}^{3d}$ as the CAM of each domain, and refer to them as 2D CAM and 3D CAM throughout this paper. For both networks, we use the GAP (Global Average Pooling) layer to average the pixel- or point-wise predictions along the spatial dimension as follows:

$$\mathbf{y}^{2d} = \frac{1}{HW} \sum_{i=1}^{H} \sum_{j=1}^{W} \mathbf{A}^{2d}_{:,i,j} \quad \text{and} \quad \mathbf{y}^{3d} = \frac{1}{N} \sum_{n=1}^{N} \mathbf{A}^{3d}_{:,n}, \tag{1}$$

where $\mathbf{y}^{2d} \in \mathbb{R}^C$ and $\mathbf{y}^{3d} \in \mathbb{R}^C$ denote image-level and scene-level class prediction, respectively. To promote the 2D and 3D CAMs to be activated on the corresponding regions of classes, the classifiers are trained with classification loss as following, where $\ell_{bce}$ denotes the binary cross entropy loss.

$$\mathcal{L}^{2d}_{cls} = \ell_{bce}(\mathbf{y}^{2d}, \mathbf{t}^{2d}) \quad \text{and} \quad \mathcal{L}^{3d}_{cls} = \ell_{bce}(\mathbf{y}^{3d}, \mathbf{t}^{3d}). \tag{2}$$

## 3.2 From 2D to 3D: Transferring semantic capability

We are inspired by an observation that the information contained in 2D and 3D data is not only distinct but also complementary. The proposed framework aims to jointly learn WSSS in both domains and unify their strengths. For joint learning of both domains, we could directly provide information from one domain to the other domain via feature paths using projection, as in [10]. However, in the inference phase, a model including the feature path requires the paired set of data of both domains, which greatly limits the practical use case.

Therefore, instead of directly swapping the features, we devise our framework to provide information from one domain to the other domain as a form of self-supervision during the training process only. It enables the proposed 2D-3D framework to separately process the data of each domain – our framework can perform 2D WSSS and 3D WSSS individually in the inference phase, without using the input data from the other domain.

We use the view matrices of images for bridging the 2D and 3D domains, as visualized in Fig. 2. The projection from 3D homogeneous coordinates to the 2D pixel coordinates in $\mathbf{I}^k$ is formulated as follows:

$$[u_n^k, v_n^k, 1]^T = \mathbf{V}^k[x_n, y_n, z_n, 1]^T, \tag{3}$$

where $(x_n, y_n, z_n)$ is the coordinate of the $n$th point in the point cloud $\mathbf{P}$ and $(u_n^k, v_n^k)$ is the destination of the projected point in $\mathbf{I}^k$. We denote the projection function with the view matrix $\mathbf{V}^k$ as $proj(\cdot, \mathbf{V}^k)$. For simplicity, we omit $k$ throughout this paper.

With a given $\mathbf{V}$, the 3D point features $\mathbf{Z}^{3d}$ and 3D CAM $\mathbf{A}^{3d}$ are projected onto the image plane. Here, we discard the points located outside of the image grid. Also, we filter the points of the occluded object which should not exist on the image. The projected 3D features $\mathbf{Z}^{3d,p} \in \mathbb{R}^{D \times N_v}$ and 3D CAM $\mathbf{A}^{3d,p} \in \mathbb{R}^{C \times N_v}$ can be obtained as follows:

$$\mathbf{Z}^{3d,p} = proj(\mathbf{Z}^{3d}, \mathbf{V}) \quad \text{and} \quad \mathbf{A}^{3d,p} = proj(\mathbf{A}^{3d}, \mathbf{V}), \tag{4}$$

where $N_v$ denotes the number of preserved points. Note that the projection process is differentiable, and therefore the gradients can be back-propagated (see blue dashed lines in Fig. 2).

Once the projected 3D CAM is acquired, we bridge the 2D CAM and 3D CAM at the pixels where the projected points exist. To leverage the semantic advantages of the 2D domain for improving 3D CAM, we obtain a pseudo-label map $\mathbf{M}^{2d}$ from the 2D CAM. Since there is usually no explicit background class in indoor scenes, we regard the class with maximum prediction score as a predicted class at each pixel as follows:

$$\mathbf{M}_{i,j}^{2d} = \underset{c \in \{1, \dots, C\}}{\operatorname{argmax}} (\mathbf{t}_c^{2d} \cdot \mathbf{A}_{c,i,j}^{2d}), \tag{5}$$

where the non-existing classes are ignored by using the given image-level class label $\mathbf{t}^{2d}$.

We apply point-wise classification loss to the 3D CAM to follow the acquired 2D pseudo-label. However, unlike GT supervision, the predicted 2D pseudo-labels may include inaccurate signals and cannot be fully trusted. To address this, we regard the maximum score of class prediction of each pixel as a metric for the confidence of that pixel. The confidence map $\mathbf{W}^{2d}$ is defined as follows:

$$\mathbf{W}_{i,j}^{2d} = \max(\mathbf{A}_{:,i,j}^{2d}). \tag{6}$$

We define the 2D-to-3D loss that transfers 2D semantic capability to 3D network as

$$\mathcal{L}_{joint}^{2d \to 3d} = \frac{1}{N_v} \sum_{n=1}^{N_v} \mathbf{W}_{u_n,v_n}^{2d} \ell_{bce}(\mathbf{A}_{:,n}^{3d,p}, \mathbf{M}_{u_n,v_n}^{2d}), \tag{7}$$

where the $(u_n, v_n)$ is a pixel coordinate of the $n$th projected point as we denoted. Using maximum logit as a confidence score can reduce the contributions of such less confident pixels to the loss, and prevent 3D CAM from learning with erroneous self-supervision. We verify that this confidence-based scoring method brings actually increases the WSSS performance by a large gap, which will be demonstrated in Section 4.

### 3.3 From 3D to 2D: Transferring segmentation capability

Thanks to the well-preserved geometrical structures of the point cloud, the 3D WSSS can be advantageous for segmenting the scenes compared with 2D WSSS. However, unlike 2D-to-3D loss utilizing 2D pseudo-labels, this benefit cannot be directly leveraged to mitigate the impreciseness of 2D CAM due to the class imbalance issue and low semantic perception capability of the 3D CAM. In this paper, to unlock the segmentation potentials of the point cloud classifier, we train the image classifier to generate 2D features having a similar correlation matrix with that of the 3D features.

Motivated by [19], we construct a correlation matrix $\mathbf{S}^{3d,p}$ from the projected 3D features $\mathbf{Z}^{3d,p}$, by using the following equation:

$$\mathbf{S}_{a,b}^{3d,p} = \mathbf{Z}_{:,a}^{3d,p} \cdot \mathbf{Z}_{:,b}^{3d,p}, \tag{8}$$

where $(a, b)$ denotes a pair of the projected points. Similarly, for the pixels corresponding to the projected points, $\mathbf{S}^{2d}$ is computed from 2D features $\mathbf{Z}^{2d}$ as follows:

$$\mathbf{S}_{a,b}^{2d} = \mathbf{Z}_{:,u_a,v_a}^{2d} \cdot \mathbf{Z}_{:,u_b,v_b}^{2d}, \tag{9}$$

where the $(u_a, v_a)$ and $(u_b, v_b)$ denote pixel coordinates of the $a$th and $b$th projected point, respectively. Note that we map the features on a unit hypersphere by normalization, before computing the correlation matrices. Similar to the 2D-to-3D loss, we construct $\mathbf{W}^{3d,p}$, the confidence of the correlation matrix of projected 3D features, by multiplying the confidence at each point as follows:

$$\mathbf{W}_{a,b}^{3d,p} = \max(\mathbf{A}_{:,a}^{3d,p}) \cdot \max(\mathbf{A}_{:,b}^{3d,p}). \tag{10}$$

The confidence-based 3D-to-2D loss is then defined as follows:

$$\mathcal{L}_{joint}^{3d \rightarrow 2d} = \frac{1}{N_v^2} \sum_{a=1}^{N_v} \sum_{b=1}^{N_v} \mathbf{W}_{a,b}^{3d,p} |\mathbf{S}_{a,b}^{2d} - \mathbf{S}_{a,b}^{3d,p}|_1. \tag{11}$$

In summary, we apply the sum of the classification and the joint loss functions for training our framework as following equation, where $\lambda$ is a weighting parameter.

$$\mathcal{L}_{total} = \mathcal{L}_{cls}^{2d} + \mathcal{L}_{cls}^{3d} + \mathcal{L}_{joint}^{2d \rightarrow 3d} + \lambda \mathcal{L}_{joint}^{3d \rightarrow 2d}. \tag{12}$$

## 4 Experiments

### 4.1 Implementation details

**Framework** The proposed framework is implemented with PyTorch. ResNet38 [30] and Point-Net++ [21] are employed as backbones for the image classifier and point cloud classifier, respectively. The image classifier is initialized with the weights pre-trained on ImageNet [23]. We augment the input point cloud with sub-sampling, random flipping, and random rotation. For the image, random resizing/cropping, horizontal flipping, and color jittering [14] are applied. We set $\lambda = 1$ in Eq. 12. The model is trained on two Tesla V100 GPUs with batch size 16 for 200 epochs. The initial learning rate is set to 0.003, and be decayed by 0.1 at epoch 120, 160, 180 as in [22]. More details can be found in *Supplementary Material*.

**Dataset and metric** Following the existing 3D WSSS studies [29, 22], we conduct experiments on ScanNetV2 [5] dataset (MIT license, we agreed to the terms of use), which contains indoor scenes such as offices or bathrooms. ScanNetV2 is composed of 1513 scans with annotations (semantic segmentation GT, camera poses, etc.) and 100 *test* scans serving as a benchmark. We follow the official split, where there exist 1201 *train* scans and 312 *val* scans. For the 2D data, we use the provided RGB frames and the corresponding camera view matrices. We sample only 1% of all provided frames, which is equivalent to using around 17 frames per scene on average. Note that the proposed label setting is less expensive than that of MPRM [29] (using around 17 class-labeled subclouds per scene), since annotating class-level labels is even easier on images than on point clouds. For evaluating the semantic segmentation performance, we use the mean intersection over union (mIoU) as a metric on both the 2D and 3D domains. Since the GT for the test set is not publicly available, *test* performance is evaluated on the ScanNet evaluation server.

Table 2: Ablation study of the proposed framework. The semantic segmentation performance (mIoU, %) is evaluated on the official *train* and *val* splits of the ScanNetV2 [5]. From left to the right, each loss term denotes image-level 2D classification loss ( $\mathcal{L}^{2d}_{cls}$ ), scene-level 3D classification loss ( $\mathcal{L}^{3d}_{cls}$ ), 2D-to-3D loss ($\mathcal{L}^{2d\to3d}_{joint}$), and 3D-to-2D loss ($\mathcal{L}^{3d\to2d}_{joint}$ ). Confidence-based scoring denotes using $\mathbf{W}$ (in Eq. 6 and Eq. 10). **Bold** numbers represent the best results.

| Domain | $\mathcal{L}^{2d}_{cls}$ | $\mathcal{L}^{3d}_{cls}$ | $\mathcal{L}^{2d\to3d}_{joint}$ | $\mathcal{L}^{3d\to2d}_{joint}$ | Confidence-based scoring | Train | Val |
|---|---|---|---|---|---|---|---|
| 2D | ✓ | | | | | 38.0 | 33.3 |
| | ✓ | ✓ | | ✓ | | 47.8 | 39.2 |
| | ✓ | ✓ | | ✓ | ✓ | 50.0 | 41.6 |
| | ✓ | ✓ | ✓ | ✓ | ✓ | **54.4** | **44.5** |
| 3D | | ✓ | | | | 17.9 | 16.2 |
| | ✓ | ✓ | ✓ | | | 51.3 | 43.8 |
| | ✓ | ✓ | ✓ | | ✓ | 53.5 | 45.6 |
| | ✓ | ✓ | ✓ | ✓ | ✓ | **59.1** | **49.6** |

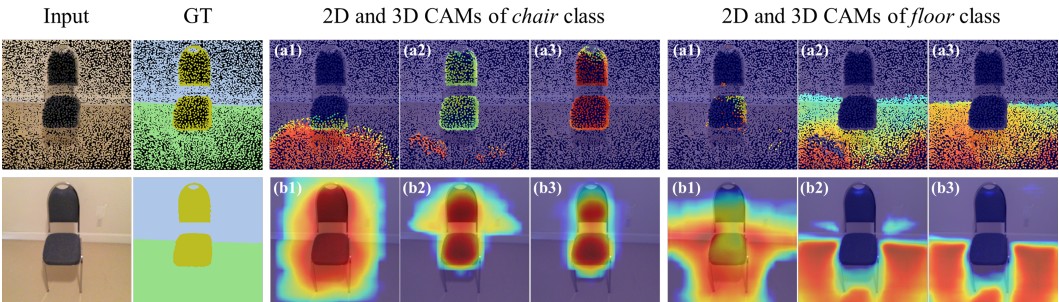

Figure 3: Qualitative comparison of 2D/3D CAMs. **3D CAMs** – (a1) uses scene-level classification loss ($\mathcal{L}^{3d}_{cls}$) only, (a2) uses 2D-to-3D loss ($\mathcal{L}^{2d\to3d}_{joint}$), and (a3) uses bidirectional (2D-to-3D and 3D-to-2D, *i.e.*, $\mathcal{L}^{2d\to3d}_{joint} + \mathcal{L}^{3d\to2d}_{joint}$) joint loss. **2D CAMs** – (b1) uses image-level classification loss ($\mathcal{L}^{2d}_{cls}$) only, (b2) uses 3D-to-2D loss ($\mathcal{L}^{3d\to2d}_{joint}$), and (b3) uses bidirectional joint loss. More qualitative comparison and samples of 2D and 3D CAMs can be found in *Supplementary Material*.

## 4.2 Ablation studies

To demonstrate the advantage of the proposed joint learning framework, we conduct an ablation study, as shown in Table 2. For each domain, the first row shows the result of our baseline setting, where only the image- or scene-level classification loss of each domain ($\mathcal{L}^{2d}_{cls}$ or $\mathcal{L}^{3d}_{cls}$) is applied. As reported in prior works [29, 22], we also observe that the 3D WSSS suffers from the class imbalance of scene-level class labels and thereby completely fails to understand the semantics of the objects, as visualized in (a1) of Fig. 3. On the other hand, as shown in (b1), 2D WSSS shows the capability to roughly localize the region of each class on the image. However, due to the lack of geometrical 3D information, the 2D CAM fails to be precise at the object boundary. This illustrates the motivation of the proposed 2D-3D joint framework.

A benefit of the **2D-to-3D loss** ($\mathcal{L}^{3d\to2d}_{joint}$) is shown in the second row of the 3D domain in Table 2. The *train* and *val* performance of 3D CAM are significantly increased from 17.9% and 16.2% to 51.3% and 43.8%, respectively. The 3D CAMs in Fig. 3 (a2) show much more accurate results with the help of the 2D-to-3D joint loss. Here, to clarify the benefit of the 2D-to-3D loss, we also conduct an additional experiment. In this experiment, instead of the joint loss, we apply an additional classification loss to the point cloud classifier by using the image-level class labels. This setting is similar to that of PCAM [29]; however, the classification is done on the image level, not on the subcloud level. It achieves 43.9% and 39.8% on *train* and *val* split respectively, which are the almost equivalent performance to PCAM (43.1% and 38.1%) but clearly lower than using our 2D-to-3D loss (51.3% and 43.8%). The result indicates that the performance gain of the proposed 2D-to-3D loss not only comes from relieving the class imbalance problem of the 3D domain but also effectively transferring the better semantics of the 2D domain to the 3D domain.

Table 3: Performance (mIoU, %) comparison with other state-of-the-art 3D WSSS methods [29, 22] on ScannetV2 [5] *train*, *val*, and *test* split. Class-wise IoUs can be found in *Supplementary Material*. **Bold** numbers represent the best results, while underlined numbers are the second best results.

| Methods | Backbones | Weak Labels | Train | Val | Test |
|---------|-----------|-------------|-------|-----|------|
| PCAM[29] | KPConv [27] | Scene | 22.1 | - | - |
| | | Subcloud | 43.1 | 38.1 | - |
| MPRM[29] | | Scene | 24.4 | - | - |
| | | Subcloud | 47.4 | 43.2 | 41.1 |
| MIL-seg[22] | PointNet++ [21] | Scene | - | 20.7 | - |
| WyPR[22] | | | 30.7 | 29.6 | 24.0 |
| proposed | PointNet++ | Scene+Image | **59.1** | **49.6** | **47.4** |

The results also show that the **3D-to-2D loss** ($\mathcal{L}_{joint}^{3d \rightarrow 2d}$) enhances the performance of 2D WSSS, from 38.0% and 33.3% to 47.8% and 39.2%, respectively. Thanks to the geometrical information provided by the 3D domain in the form of a correlation matrix, as visualized in Fig. 3 (b2), the generated 2D CAMs are much more precise at the object boundary when being trained with the 3D-to-2D loss. The results strongly support our design intentions: the 2D-to-3D loss provides rich semantics from 2D to 3D, while the 3D-to-2D loss transfers the segmentation capability of 3D to 2D.

Further, we quantitatively verify that the confidence-based scoring strategy is effective in both the 2D-to-3D and 3D-to-2D directions, by suppressing the self-supervision at inaccurate regions while preserving reliable guidance. Finally, by using the bidirectional (2D-to-3D and 3D-to-2D) joint losses, we even further improve the semantic segmentation performance in both domains (the fourth row of each domain in Table 2 and (a3)/(b3) in Fig. 3). It shows the superiority of the proposed joint framework, which unlocks the complementary potential of 2D and 3D domains and unifies them.

Here, we can observe that the performance of 2D WSSS is quite lower than that of 3D WSSS after we apply joint losses. One of the main reasons is the limited FoV (Field of View) of the perspective camera that captures the images. When only the corner of an object is captured in the image, it is almost impossible to correctly perceive the object. Therefore, classification results and corresponding 2D CAMs could be also inaccurate. To relieve the issue, during the training process, we filter 2D CAM with ground truth image-level class labels and suppress confusing regions. However, misclassification that occured in the inference phase is difficult to be addressed in our framework. Extending the proposed framework using the cameras with large FoV (*e.g.* omnidirectional camera) can be an interesting future direction.

## 4.3 Segmentation results

In Table. 3, we compare the point cloud semantic segmentation performance of the proposed framework with that of the other 3D WSSS methods [29, 22]. This shows the superiority of the proposed 2D-3D joint framework, surpassing the other methods by more than 6% in every split. Although we use additional 2D image data and image-level class labels compared to WyPR [22], annotating additional image-level class labels is not very expensive, particularly when we consider the significant performance gain. Compared with MPRM [29] using subcloud-level labels, the annotating cost is rather inexpensive at the image-level compared to the subcloud-level, while the proposed framework outperforms MPRM by a significant margin. It is also noteworthy that the proposed framework does not employ the two-stage retraining technique or dense conditional random field (CRF), unlike MPRM. In summary, the proposed 2D-3D joint learning method outperforms the existing 3D WSSS methods in both terms of label efficiency and performance.

Figure 4 visualizes qualitative comparisons of the proposed framework and baselines, in both 2D and 3D domains. Compared with the result learned with a single 2D or 3D domain, our framework shows significantly better results in terms of accuracy of semantics and preciseness of segmentation. With our 2D-3D joint framework for WSSS, the complementary advantages of both domains are effectively combined. We can observe that the unified benefits of multi-domains achieve substantial semantic segmentation results, compared to uni-domain baselines. There are a few limitations such as thin objects (*e.g.* legs of a table) in the 3D domain, fine details in the 2D domain, and sensitivity to the pose quality. Nevertheless, our framework achieves a strong baseline for the 2D-3D joint learning of WSSS. Additional semantic segmentation results are provided in *Supplementary Material*.

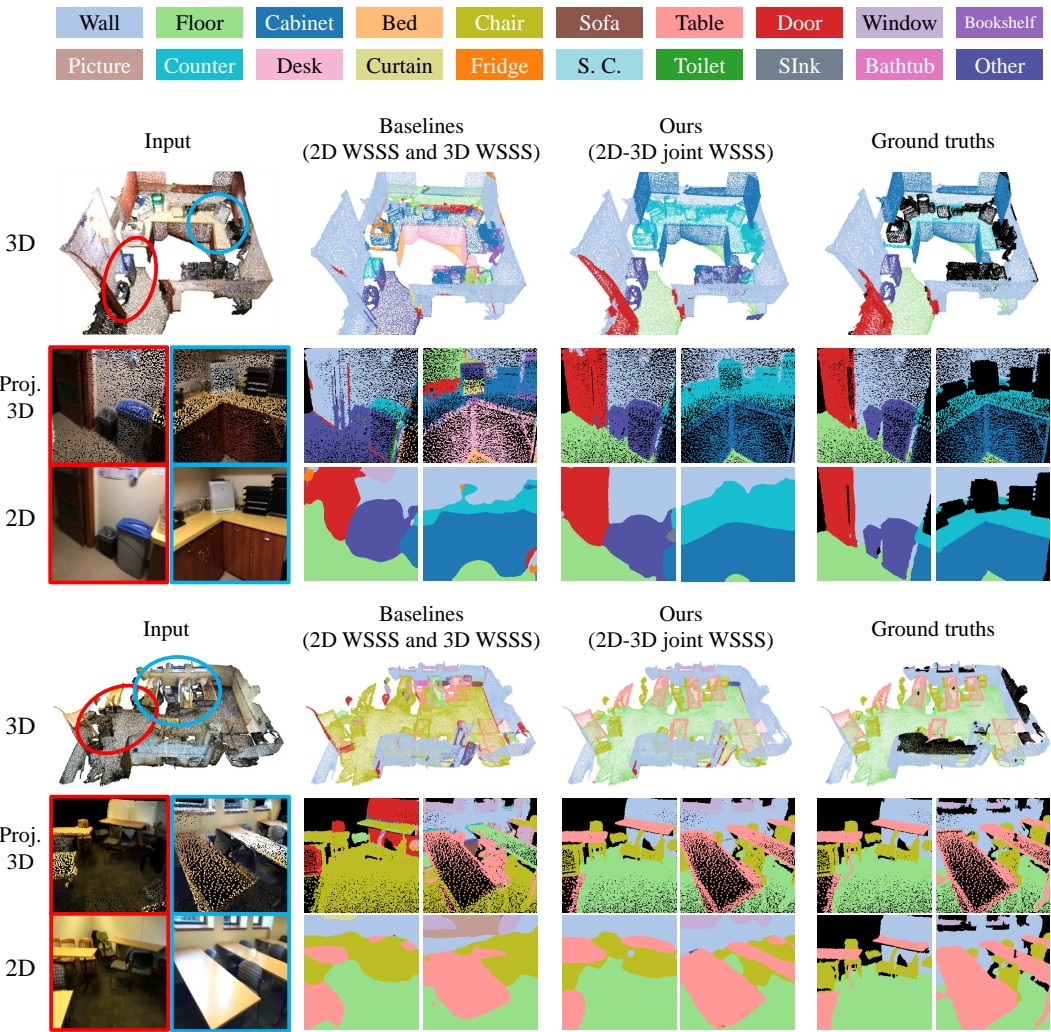

| Wall | Floor | Cabinet | Bed | Chair | Sofa | Table | Door | Window | Bookshelf |
|---|---|---|---|---|---|---|---|---|---|
| Picture | Counter | Desk | Curtain | Fridge | S. C. | Toilet | SInk | Bathtub | Other |

Figure 4: Qualitative comparison of 2D/3D semantic segmentation results of uni-domain baselines and the proposed 2D-3D joint framework. From left to right: input (projected) point clouds and images, results of uni-domain 2D WSSS and 3D WSSS, our joint WSSS results, and ground truths. We visualize the region of the point cloud corresponding to the images by red or blue circles. A color map of the 20 classes of ScanNetV2 [5] is provided at the top of the figure, where the black color denotes the ignore index.

## 4.4 Generalization Capability

Most of the existing (indoor) point cloud datasets [5, 2] have obtained the point cloud from a sequence of RGBD frames, using 3D reconstruction techniques such as Structure from Motion (SfM). Therefore, in addition to the 3D point clouds, such datasets usually provide 2D images and the poses of the 2D images. The proposed joint learning framework targets such paired multi-domain datasets. Throughout Sec 4, we demonstrated that unifying the complementary advantages of multi-domain data could bring remarkable improvements in both 2D and 3D WSSS.

Nevertheless, we agree that the paired multi-domain data is not trivial to acquire in usual, especially when we want to apply our framework to the novel data of the unobserved scene domains. Compared to the existing 2D or 3D WSSS methods which require single-domain data (*e.g.* images only or point cloud only), the difficulty to acquire the multi-domain dataset is one of the limitations of the proposed joint WSSS framework.

Table 4: Quantitative comparisons of the generalization capability between the proposed joint learning framework and our baselines. The semantic segmentation performance (mIoU, %) is evaluated on the official *val* splits of the S3DIS [2] dataset. The loss terms follow the denotation of Table 2. We use confidence-based scoring by default. **Bold** numbers represent the best results.

| Domain | $\mathcal{L}_{cls}^{2d}$ | $\mathcal{L}_{cls}^{3d}$ | $\mathcal{L}_{joint}^{2d\to3d}$ | $\mathcal{L}_{joint}^{3d\to2d}$ | Val |
|---|---|---|---|---|---|
| | ✓ | | | | 36.9 |
| 2D | ✓ | ✓ | | ✓ | 43.3 |
| | ✓ | ✓ | ✓ | ✓ | **49.5** |
| | | ✓ | | | 17.5 |
| 3D | ✓ | ✓ | ✓ | | 48.0 |
| | ✓ | ✓ | ✓ | ✓ | **54.4** |

To relieve this issue, we experimentally verify the generalization capability of the proposed method using the S3DIS [2] dataset, as in WyPR [22]. In this experiment, we first train our model on the ScanNetV2 dataset only and then test the trained model on validation set of the S3DIS dataset. We evaluate the performance only on the overlapped class between the ScanNetV2 and S3DIS datasets. Here, it is noteworthy that we perform 2D WSSS and 3D WSSS individually, without using the input data from the other domain. Thanks to our practical design philosophy, we could feed the 3D point clouds only for the 3D branch and 2D images only for the 2D.

Table 4 shows the generalization capability of our method by comparing it with our baselines. Similar to the tendency in Table 2, the proposed 2D-3D joint loss provides significant performance gain in both 2D and 3D domains, even on the dataset unobserved during the training. Especially, the proposed framework still outperforms the 3D WSSS performance of WyPR [22] on the S3DIS dataset (22.3%) by a large gap. The results strongly support that the complementary advantages of 2D and 3D domains are preserved on the other dataset, even without any further fine-tuning or domain-specific adaptation.

Although the proposed joint framework requires paired multi-domain data during training, thanks to the demonstrated generalization capability, our framework could be scaled into processing the unobserved general data. From a practical perspective, this generalizability of the proposed method is a strong contribution to the field of WSSS.

## 5   Conclusion

WSSS has been widely studied for relieving the annotation cost of semantic segmentation in both 2D and 3D domains. However, the existing studies have been focused on domain-specific methods. In this paper, we propose a novel 2D-3D joint framework for WSSS in both 2D and 3D domains. Motivated by the complementary nature of the 2D and 3D domains, we devise our framework to effectively transfer the benefit of one domain to the other domain in a simultaneous manner. We bridge the 2D pixels and 3D points via projection and leverage the 2D CAM as self-supervision for improving the semantic perception of the 3D CAM. To transfer the geometrical structure of the 3D scene to the 2D domain, we exploit a correlation matrix of 3D features for training the 2D features. For both directions, we propose a confidence-based scoring strategy to suppress less accurate self-supervision and make the framework learn from the reliable regions only. With extensive experiments, we verify that the proposed 2D-to-3D and 3D-to-2D joint loss effectively unifies the distinct benefits of both domains, while relieving the disadvantages. On the ScanNetV2 dataset, the proposed framework significantly outperforms the single-domain baselines and the other state-of-the-art methods. Further, with an experiment on the S3DIS dataset, we demonstrate the generalization capability of our method, which supports its practicality. We believe that 2D-3D joint learning is a new and promising research direction for WSSS, and this paper provides a strong baseline for it.

## Acknowledgments and Disclosure of Funding

This work was supported by the National Research Foundation of Korea(NRF) grant funded by the Korea government(MSIT) (NRF2022R1A2B5B03002636).

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
