# Supplementary Material: Joint Learning of 2D-3D Weakly Supervised Semantic Segmentation

**Hyeokjun Kweon**
KAIST
0327june@kaist.ac.kr

**Kuk-Jin Yoon**
KAIST
kjyoon@kaist.ac.kr

## 1 Implementation details

### 1.1 Training details

In the proposed framework, we employ ResNet38 [7] and PointNet++ [4] as an image classifier and a point cloud classifier, respectively. As we introduced in Section 3 of the main paper, we add a linear layer for acquiring pixel- or point-wise class prediction from the features. For training the image classifier, we use a poly learning rate which multiplies $(1 - \frac{iter}{max\ iter})^{power}$ to the initial learning rate. As in [3]. we set the initial learning rate and the $power$ as 0.01 and 0.9, respectively. In the training process, we sample one image per point cloud and bridge them with the help of the corresponding view matrix. We also tested using multiple images at once; however, the performance gain was marginal when we consider the increase in memory requirements and training times.

In the first phase, we individually train both 2D and 3D classifiers with the classification loss of each domain. After that, we jointly train them using the proposed 2D-to-3D and 3D-to-2D losses, in addition to the classification loss. Here, in the second phase, please note that we first train our framework without the 3D-to-2D loss for the first few epochs. Although we pre-trained the 3D network in the first phase and thereby it can extract meaningful 3D features for segmentation to some degree, it is true that the 2D network shows much better semantic segmentation capability in the first phase (refer to the 2D and 3D baselines in Table 4 of the main paper). Since this imbalance might lead to unstable joint training, we strengthen the 3D network with 2D-to-3D loss in the early epochs of the second phase.

### 1.2 Data augmentations

As we explained in Section 4.1 of the main paper, we augment the images and point clouds before we feed them to our framework. In this subsection, we provide our augmentation methods in detail.

**Image augmentation**  We first re-scale the longer axis of the input image to a random size in the range of [256, 512], and crop a 256×256 size patch from the resized image. Here, we empirically observe that using a bigger patch size is ineffective in terms of classification, since having a wider relative receptive field is crucial for understanding the scene from the image. On the other hand, when we use a smaller patch size, fine details of the image could not be preserved. After that, we randomly apply horizontal flipping ($p = 0.5$), and adjust the color distribution of the image by using color jittering [2] with the following parameters: brightness=0.3, contrast=0.3, saturation=0.3, and hue=0.1. We find that color jittering is essential since the frames of ScanNetV2 [1] have a wide variety in terms of brightness and saturation. We also observe that some frames are spoiled by severe motion blurs. Enhancing such frames with off-the-shelf modules (*e.g.* deblurring network) could be effective for improving our framework.

**Point cloud augmentation**  For augmenting the input point cloud, we sub-sample 40k points from the original point cloud. When there are less than 40k points, we allow repetition. We apply random

36th Conference on Neural Information Processing Systems (NeurIPS 2022).

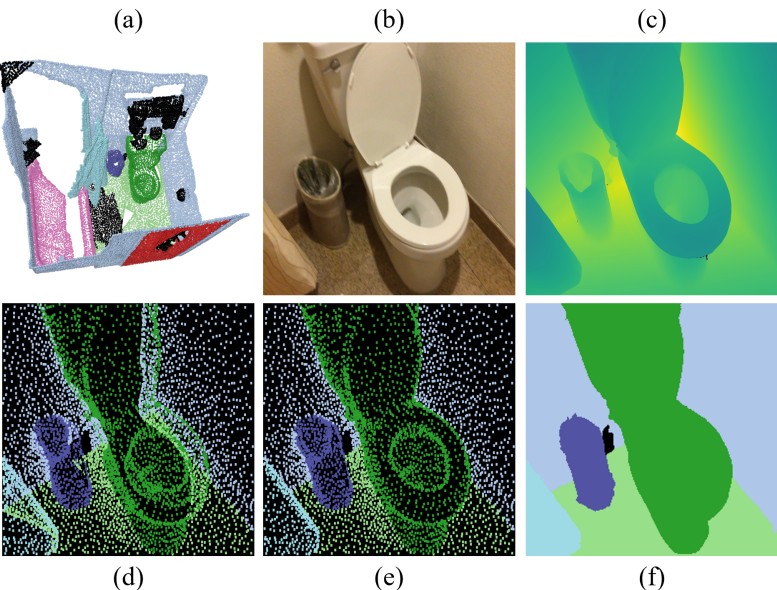

(a)             (b)             (c)

(d)             (e)             (f)

Figure 1: Comparison between the results of projection with and without the depth-based filtering. From (a) to (f): a point cloud, an image, depth map corresponding to the image, projected 3D GT without filtering, projected 3D GT with filtering, and 2D semantic segmentation GT of the image.

flipping ($p = 0.5$) to the sampled point cloud in both horizontal and vertical directions. We also adjust the point cloud by using random rotation of [-5,5] degrees around the upright axis. Note that we leverage axis alignment information given from the ScanNetV2 [1] dataset for *train* and *val* sets, as in WyPR [5].

## 2   Details on rendering

With a given view matrix, we project the 3D features and 3D CAMs to the image plane. Here, as explained in Section 3.2 of the main paper, we discard the points located outside of the image grid. Also, we filter the points of the occluded object which should not exist on the image. For filtering the occlusion, we exploit the depth map corresponding to the image provided by the ScanNetV2 dataset as a reference. We could also render a depth map from the mesh (generated from the point cloud). Then, the points having the same depth as the reference depth are preserved during projection, while the other points are filtered. With the filtering process, we can safely reject the occluded points that should not be trained by 2D CAMs. Figure 1 compares the results of projection with and without the filtering. In the figure, for better understanding, we visualize the projection results of ground truth (GT) semantic segmentation labels of the point cloud. Since the projection is done by matrix-vector multiplication, the overall rendering process is differentiable. Thereby the gradients obtained from the 2D-to-3D loss ($\mathcal{L}_{joint}^{2d \rightarrow 3d}$) between the projected 3D CAM and 2D pseudo-label can be back-propagated to the point cloud classifier, as visualized in the Fig. 2 of the main paper.

## 3   Semantic segmentation results

In Table 1, we show the class-wise segmentation performance of the proposed framework and the other 3D WSSS studies [5, 6]. We can observe that our framework outperforms all of the existing methods by a large gap in both *train* and *val* sets. Even though we did not employ refinement techniques such as denseCRF or retraining used in MPRM [6], the proposed method achieves remarkable performance with the help of the proposed 2D-3D joint learning. We also provide the 2D class-wise segmentation performance of the proposed framework and the 2D baseline (2D WSSS) in Table 2. We can observe that our framework outperforms the baseline, thanks to the segmentation capability transferred from the 3D domain via $\mathcal{L}_{joint}^{3d \rightarrow 2d}$. The results support the superiority of the proposed framework, which successively unifies the benefit of WSSS in both 2D and 3D domains.

Table 1: Class-wise IoU on ScanNetV2 [1] *train* and *val* split. We compare the results of the proposed framework with those of the existing 3D WSSS works [5, 6]. Here, "refine" denotes refinement techniques (denseCRF or retraining) employed by MPRM [6]. For simplicity, we abbreviate *cabinet/window/bookshelf/picture/counter/curtain/shower curtain/other furniture* as cab./win./B.S./pic./cnt./cur./S.C./O.F., respectively. **Bold** numbers represent the best results.

| Method | split | refine | wall | floor | cab. | bed | chair | sofa | table | door | win. | B.S. | pic. | cnt. | desk | cur. | fridge | S.C. | toilet | sink | tub | O.F. | mIoU |
|---|---|---|---|---|---|---|---|---|---|---|---|---|---|---|---|---|---|---|---|---|---|---|---|
| WyPR [5] | train | No | 59.3 | 31.5 | 6.4 | 58.3 | 31.6 | 47.5 | 18.3 | 17.9 | 36.7 | 34.1 | 6.2 | 36.1 | 24.3 | 67.2 | 8.7 | 38.0 | 17.9 | 28.9 | 35.9 | 8.2 | 30.7 |
| MPRM [6] | train | No | 56.1 | 54.8 | 32.0 | 69.6 | 49.5 | 67.7 | 46.6 | 41.3 | 44.2 | 71.5 | 28.3 | 21.3 | 49.2 | 71.8 | 38.1 | 42.8 | 43.6 | 20.3 | 49.0 | 33.8 | 46.6 |
| MPRM [6] | train | Yes | 58.0 | 57.3 | **33.2** | **71.8** | 50.4 | 69.8 | 47.9 | **42.1** | 44.9 | 73.8 | **28.0** | 21.5 | 49.5 | 72.0 | 38.8 | 44.1 | 42.4 | 20.0 | 48.7 | 34.4 | 47.4 |
| Ours | train | No | **72.6** | **89.7** | 32.7 | 71.3 | **70.0** | **73.0** | **52.9** | 38.6 | **56.0** | **77.0** | 17.4 | **42.4** | **53.8** | **74.6** | **49.5** | **74.9** | **80.0** | **36.5** | **77.4** | **42.7** | **59.1** |
| WyPR [5] | val | No | 58.1 | 33.9 | 5.6 | 56.6 | 29.1 | 45.5 | 19.3 | 15.2 | 34.2 | 33.7 | 6.8 | 33.3 | 22.1 | 65.6 | 6.6 | 36.3 | 18.6 | 24.5 | 39.8 | 6.6 | 29.6 |
| MPRM [6] | val | No | 55.7 | 50.7 | 23.1 | 57.5 | 47.5 | 53.5 | 39.2 | 32.6 | 41.8 | 63.6 | **19.7** | 19.2 | 39.8 | 66.3 | 22.2 | **44.1** | 49.1 | 23.4 | 43.0 | 28.5 | 41.0 |
| MPRM [6] | val | Yes | 59.4 | 59.6 | 25.1 | **64.1** | 55.7 | 58.7 | 45.6 | **36.4** | 40.3 | **67.0** | 16.1 | 22.6 | 42.9 | **66.9** | 24.1 | 39.6 | 47.0 | 21.2 | 44.7 | 28.0 | 43.2 |
| Ours | val | No | **69.6** | **90.0** | **27.9** | 61.0 | **68.7** | **62.7** | **52.3** | 34.1 | **42.0** | 65.2 | 5.8 | **42.6** | **44.4** | 60.4 | **25.3** | 33.5 | **70.9** | **38.6** | **66.5** | **31.4** | **49.6** |

Table 2: 2D Class-wise IoU on ScanNetV2 [1] frames in *train* and *val* split. We compare the results of the proposed framework with the 2D-only WSSS baseline. For simplicity, we abbreviate *cabinet/window/bookshelf/picture/counter/curtain/shower curtain/other furniture* as cab./win./B.S./pic./cnt./cur./S.C./O.F., respectively. **Bold** numbers represent the best results.

| Method | split | wall | floor | cab. | bed | chair | sofa | table | door | win. | B.S. | pic. | cnt. | desk | cur. | fridge | S.C. | toilet | sink | tub | O.F. | mIoU |
|---|---|---|---|---|---|---|---|---|---|---|---|---|---|---|---|---|---|---|---|---|---|---|
| 2D WSSS | train | 52.3 | 59.6 | 32.5 | 59.6 | 43.3 | 48.1 | 41.3 | 34.4 | 36.2 | 42.8 | **25.7** | 5.7 | 30.6 | 39.0 | 30.6 | 38.0 | 43.1 | 29.5 | 48.1 | 20.2 | 38.0 |
| Ours | train | **68.8** | **82.1** | **41.8** | **73.6** | **58.1** | **62.1** | **57.9** | **43.7** | **51.9** | **58.4** | 20.8 | **36.0** | **49.5** | **62.9** | **44.4** | **68.7** | **69.5** | **33.4** | **69.7** | **35.6** | **54.4** |
| 2D WSSS | val | 52.9 | 57.6 | 28.2 | 53.6 | 40.8 | 45.2 | 39.7 | 29.4 | 31.2 | 42.4 | **22.4** | 3.7 | 26.1 | 31.2 | 25.8 | 16.4 | 37.2 | 27.3 | 34.0 | 20.7 | 33.3 |
| Ours | val | **68.2** | **78.8** | **36.0** | **60.3** | **52.5** | **47.1** | **52.8** | **37.9** | **40.3** | **49.4** | 11.9 | **25.7** | **38.4** | **43.2** | **33.2** | **41.6** | **58.4** | **33.2** | **51.5** | **30.5** | **44.5** |

Figure 2 shows comparison between the CAMs of uni-domain baselines and the proposed framework in both 2D and 3D domains. We also provide example of 3D CAM in Fig. 3. Figure 4 and Figure 5 show more semantic segmentation results of the proposed framework in both 2D and 3D domains. Compared to the uni-domain baselines (2D WSSS and 3D WSSS), we can observe that the proposed framework achieves much more accurate and precise results, in both terms of CAMs and semantic segmentation results. The results strongly support that our method successfully unifies the benefits of the 2D and 3D domain as we intended.

## 4 Experiments regarding the class imbalance

In the field of (indoor) point cloud segmentation, a class imbalance is one of the most hindering factors. As we mentioned in line 79 of the main paper, some classes such as wall (97.3%) or floor (99.3%) commonly exist in most of the point clouds. One can also find the class frequency of the ScanNetV2 dataset in Table 1 of the MPRM [6].

For fully supervised cases, re-weighting based on class frequency could be helpful, as in many other similar tasks. On the other hand, in the case of WSSS, the severe class imbalance of ScanNetV2 dataset is difficult to be addressed with re-weighting only. For example, only 0.7% of the point cloud samples do not include the floor category. In such a challenging case, it is hard to expect the network to find meaningful classification cues, and thereby the quality of CAM is also poor.

We experimentally verified the efficacy of frequency-based re-weighting. In this experiment, we train the 3D network with re-weighted classification loss only. We observe that the re-weighting strategy does not make a meaningful improvement as shown in the Table 3.

Table 3: Class-wise IoU on ScanNetV2 [1] *train* split. We compare 3D baseline while ablating the frequency-based re-weighting strategy. For simplicity, we abbreviate the name of each category.

| Reweight | wall | floor | cab. | bed | chair | sofa | table | door | win. | B.S. | pic. | cnt. | desk | cur. | fridge | S.C. | toilet | sink | tub | O.F. | mIoU |
|---|---|---|---|---|---|---|---|---|---|---|---|---|---|---|---|---|---|---|---|---|---|
| No | 47.6 | 13.7 | 10.6 | 38.7 | 0.7 | 53.2 | 14.5 | 5.2 | 20.8 | 30.2 | 6.2 | 5.8 | 22.4 | 27.5 | 0.9 | 20.2 | 2.9 | 5.8 | 24.1 | 1.6 | 17.9 |
| Yes | 57.3 | 12.5 | 13.1 | 37.0 | 6.6 | 41.2 | 19.5 | 15.0 | 23.9 | 26.8 | 0.1 | 7.4 | 10.8 | 37.9 | 2.3 | 4.3 | 10.4 | 2.2 | 20.7 | 8.9 | 17.9 |

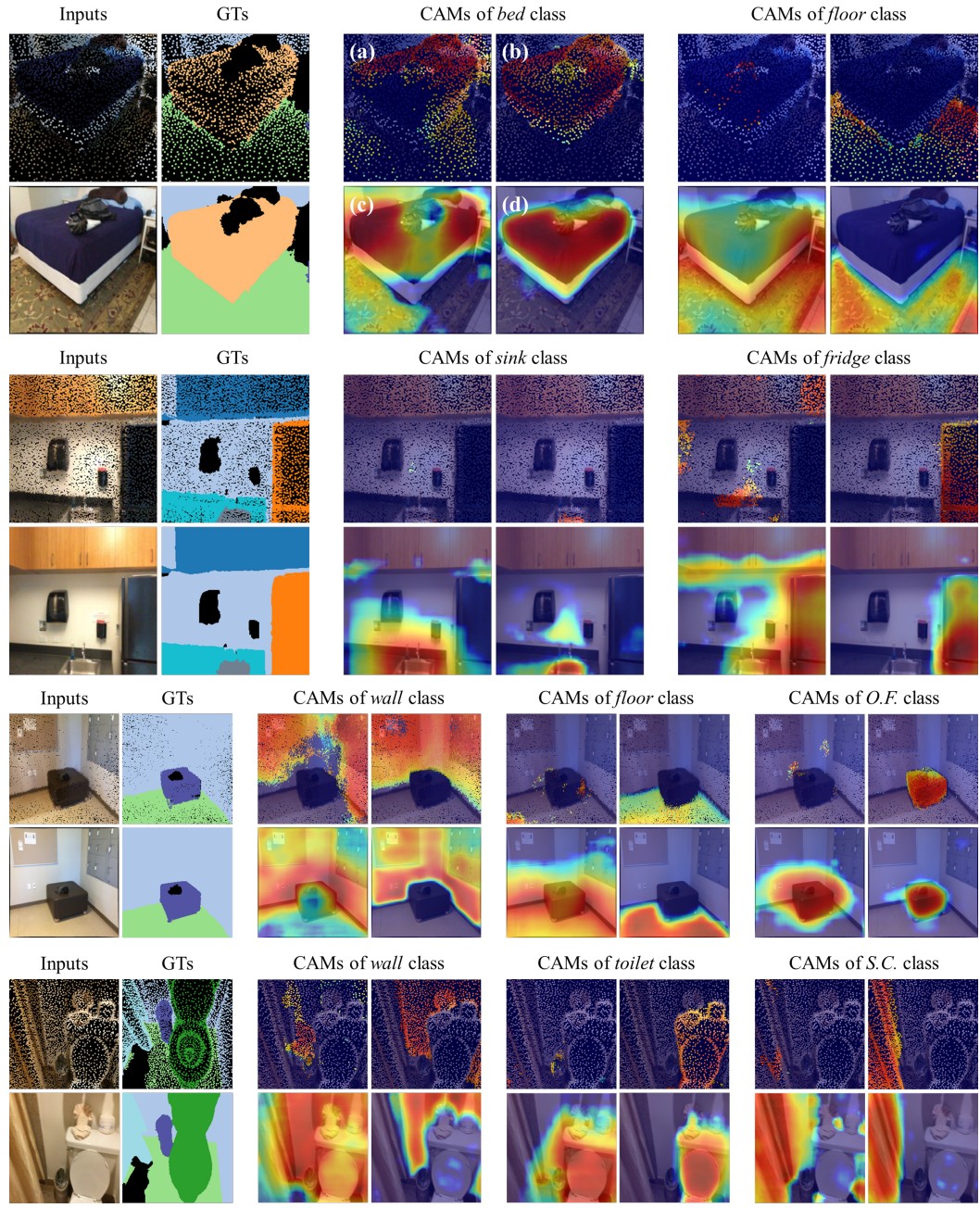

Figure 2: Comparisons of 2D and 3D CAMs of the baselines and the proposed framework. For each block of CAMs, from (a) to (d): 3D CAM of 3D WSSS baseline, ours 3D CAM, 2D CAM of 2D WSSS baseline, and ours 2D CAM. Color code follows that of the ScanNetV2 [1] dataset.

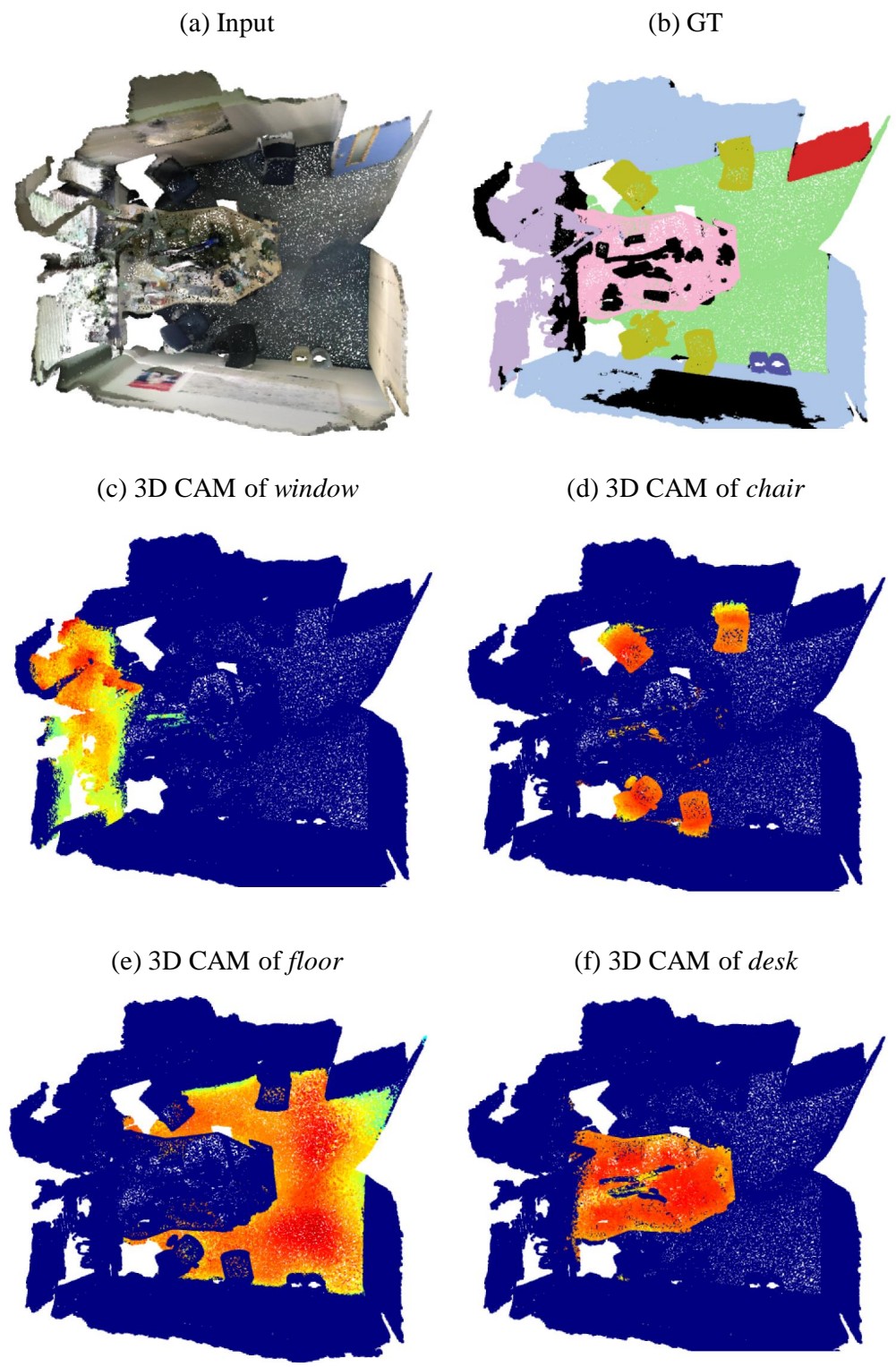

(a) Input          (b) GT

(c) 3D CAM of *window*        (d) 3D CAM of *chair*

(e) 3D CAM of *floor*         (f) 3D CAM of *desk*

Figure 3: Example of 3D CAMs. We can observe that our 2D-3D joint framework well localizes the region of each class.

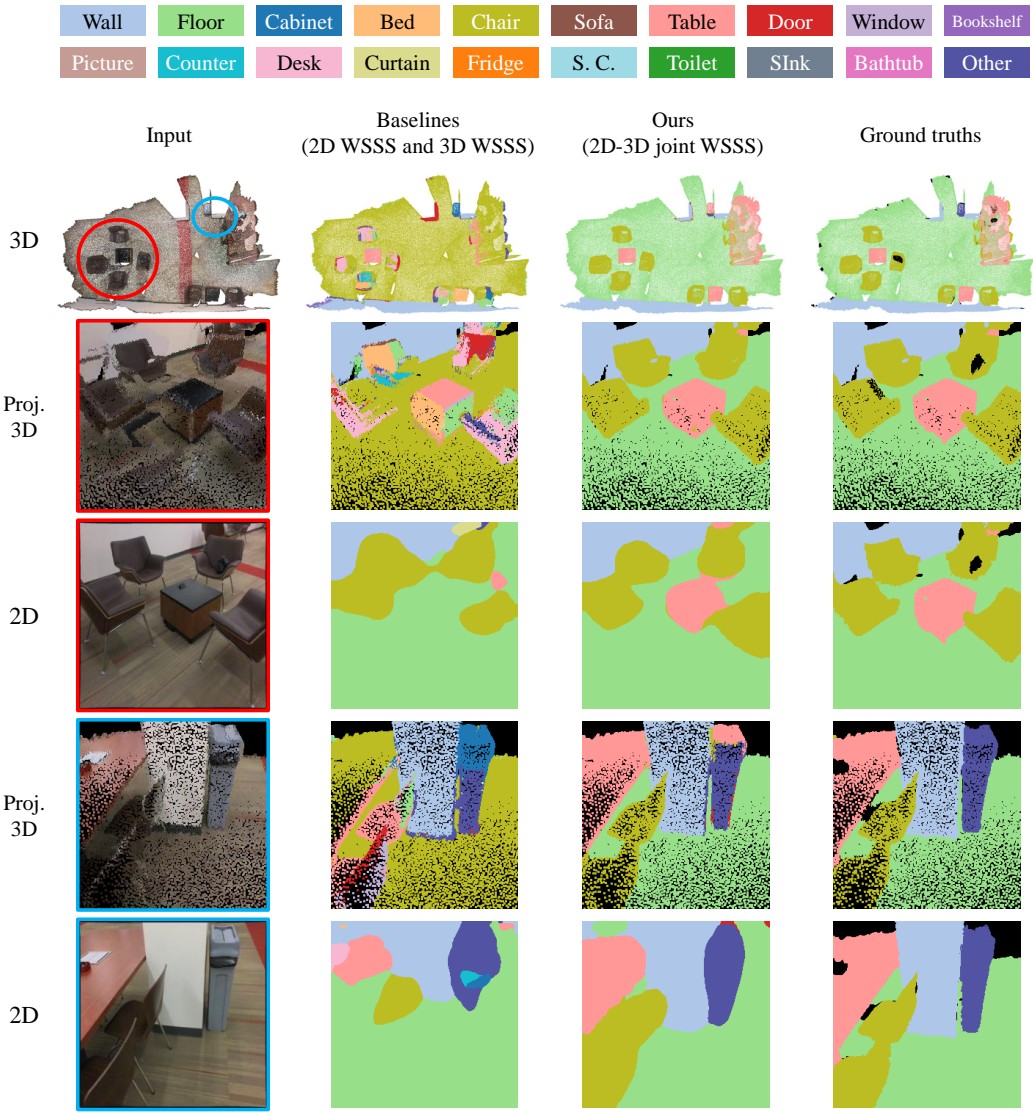

Figure 4: Qualitative comparison of 2D/3D semantic segmentation results of uni-domain baselines and the proposed 2D-3D joint framework. From left to right: input point clouds and images, results of uni-domain 2D WSSS and 3D WSSS, our results, and ground truths. A color map of the 20 classes of ScanNetV2 [1] is provided at the top of the figure, where the black color denotes the ignore index.

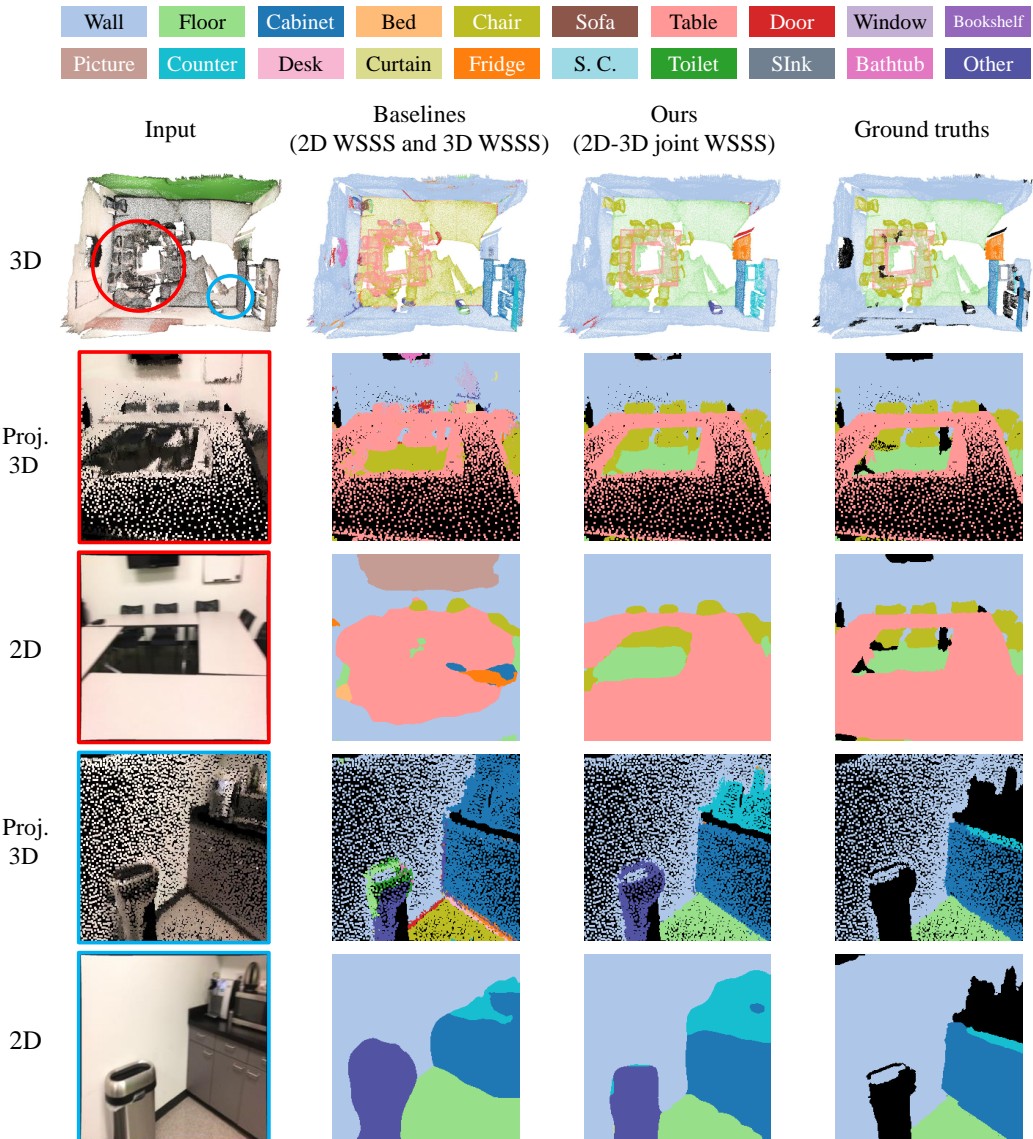

Figure 5: Qualitative comparison of 2D/3D semantic segmentation results of uni-domain baselines and the proposed 2D-3D joint framework. From left to right: input point clouds and images, results of uni-domain 2D WSSS and 3D WSSS, our results, and ground truths. A color map of the 20 classes of ScanNetV2 [1] is provided at the top of the figure, where the black color denotes the ignore index.