# OpenReview forum: "Joint Learning of 2D-3D Weakly Supervised Semantic Segmentation"
_NeurIPS.cc/2022/Conference — NeurIPS 2022 Accept_

### Official Review · Reviewer_BnBD · 2022-07-09

**Rating:** 6
**Confidence:** 3
**Soundness:** 3 good
**Presentation:** 2 fair
**Contribution:** 4 excellent

**Summary:**

The paper proposes a new task formulation and a method to jointly train semantic segmentation on images and corresponding point clouds, which can be deployed to segment any of those modalities separately. It only requires image- and scan-level class labels during training but can produce pixel- or point-level segmentation, depending on the input modality. The method makes good use of a strong semantic signal of RGB images and strong geometric signal from point clouds to disambiguate weak supervision. To achieve that, the authors introduce new loss functions to match the pre-pooling activations and the statistics on intermediate features of pixels and projected features of 3D points. That leads to significant improvement in SOTA in both 2D and 3D segmentation tasks.

**Questions:**

Please address the following, more important, concerns:
* please explain how occlusions and collisions are handled when projecting 3D features,
* please clarify the training schedule w.r.t. classification and transfer losses;
* comment on ablating pairwise correlation (instead of the direct difference of features).

**Limitations:**

Limitations are only briefly mentioned. Something to discuss:
* how sensitive is the method to the quality of camera poses used for re-projection?
* how sensitive is the method to class imbalance; how well does it work on rare classes? Maybe report a class-weighted metric (not weighted).

**Strengths And Weaknesses:**

The paper proposes an important new task formulation and a simple but efficient method to solve it, hence I vote to accept the paper, although not very confidently as WSSS is not my area.

Significance / originality:
(+) the paper proposes a practical task formulation: datasets often have both photometric and LiDAR modalities, and it is reasonable to leverage the synergy between them in absence of strong supervision;
(+) the paper appears to be the first to formulate this specific task; the proposed method is simple but not trivial;
(+) the method relaxes test-time assumptions: it can be deployed when either images or scans are available;
(−) the paper often refer to class imbalance as a hindering factor in point cloud segmentation (e.g. l. 159): can it be solved in a simpler way with re-weighting errors depending on class frequency?

Clarity:
* (±) it is possible to understand the method, although wording can be improved in multiple cases;
* (−) occlusion handling seems to be crucial when projecting 3D features / CAM, yet the paper mentions it only briefly in l. 137; are points assumed to be disks for the purpose of culling? what radius is used? Also, the paper sometimes refers to that process as rendering – that would probably require splatting the points to cover all pixels; the described process is more like projecting and rounding;
* (−) in projecting 3D features, the paper does not specify how collisions are handled, i.e. when multiple points project to the same pixel;
* (−) unclear what the training procedure is: is (12) minimised with SGD directly, or are classifiers first pre-trained and then fine-tuned with transfer losses?

Experiments:
* (+) improvement over SOTA is significant;
* (+) ablation study is sufficient: the paper ablates almost all design choices;
* (−) the paper motivates using correlation matching by CAM loss not working well; it would be good to show it experimentally, i.e. what if 3D-to-2D loss i) just uses L1 distance on features directly, not on their pairwise correlations; ii) uses it on CAMs, not Z features.

===========

Typos / wording:
* l. 19: “studies on semantic segmentation have achieved remarkable performances”;
* l. 41: “It enables our framework can individually perform...”
* in (3), is it \propto instead of the equality?
* l.153: “using maximum logit as a confidence score”: this contradicts (6) where softmax is applied to logits before taking max.

---

> ### Author Response · Authors · 2022-08-02
> **Response to Reviewer BnBD**
>
> Thank you for your thoughtful and constructive comments. We revised our paper and supplementary materials and uploaded them to OpenReview. Below we address the comments in the review point by point.
>
> ### Class imbalance
>
> * In the field of (indoor) point cloud segmentation, a class imbalance is one of the most hindering factors. As we mentioned in line 79 of the main paper, some classes such as wall (97.3%) or floor (99.3%) commonly exist in most of the point clouds. One can also find the class frequency of the ScanNetV2 dataset in Table 1 of the MPRM [29].
>
> * For fully supervised cases, we agree that re-weighting based on class frequency could be helpful, as in many other similar tasks. On the other hand, in the case of WSSS, the severe class imbalance of ScanNetV2 dataset is difficult to be addressed with re-weighting only. For example, **only 0.7% of the point cloud samples do not include the floor category**. In such a challenging case, it is hard to expect the network to find meaningful classification cues, and thereby the quality of CAM is also poor.
>
> * For more clarity, we experimentally verified the efficacy of frequency-based re-weighting. In this experiment, we train the 3D network with re-weighted classification loss only. We observe that the re-weighting strategy does not make a meaningful improvement. Class-wise IoUs of the results are provided in Table 5 of the Supplementary Material.
>
> * Also, Tables 1 and 2 in Supplementary Materials provide the class-wise IoUs of our final results in 2D and 3D domains, respectively.
>
> ### Occlusion filtering
> * At the time of submission, due to the limited space, we put the explanation about the filtering process in the Supplementary Material. Nevertheless, we humbly admit to the reviewers' comment that the mention in the main paper (line 137) is too vague. In the revision, we elaborated on the part regarding the filtering process, with the following details.
>
> * For filtering the occlusion, we exploit the depth map corresponding to the image provided by the ScanNetV2 dataset as a reference. We could also render a depth map from the mesh (generated from the point cloud). Then, the points having the same depth as the reference depth are preserved during projection, while the other points are filtered.
>
> * Also, we thoroughly agree that the name of the process should be "projecting and rounding", not a misleading "rendering".  We added the aforementioned details in the main paper. Thank you again for the constructive comment.
>
> ### Training procedure
> * We sincerely apologize for the unclear explanation of the training procedure. We revised the paper to include the following details.
> * In the first phase, we individually train both 2D and 3D classifiers with the classification loss of each domain. After that, we jointly train them using the proposed 2D-to-3D and 3D-to-2D losses, in addition to the classification loss.
>
> ### Correlation matching loss
> * Thank you for your constructive comments. We conducted an additional ablation study with the suggested settings. In the final version, we will endeavor to revise our paper to include the following details.
> * When we directly optimize the 2D features to follow the 3D features by minimizing the L1 distance between them, we observe that the training becomes severely unstable. In fact, our framework failed to converge in this setting. We think that this instability comes from the difference in features of the 2D and the 3D domains.
> * On the other hand, we also attempted to optimize the correlation matrix of CAMs itself, instead of features. In this setting, our framework achieves 53.2% (3D) and 39.1% (2D) mIoU on ScanNetV2 train split. Compared to Table 2 in the main paper, the results are much lower than those of the proposed setting in both domains.
> * In summary, the results of the ablation study show that optimizing the correlation map between the features is a more effective and feasible approach for joint WSSS.
>
> ### Sensitivity to the quality of camera poses
> * This is an interesting comment. We agree that our method could be sensitive to the quality of pose estimation. To verify the sensitivity, we put random disturbance to the perspective camera matrices while training. In specific, we add standardized Gaussian noises to the camera parameters to mimic the real disturbance.
> * As we expected, the performance is non-negligibly degraded in both 2D and 3D domains. Although we could not conduct an extensive experiment regarding the noise due to the limited time, we added it as a limitation of our framework in the revision (line 253-254). We will try to add some more experiments for it in the final version.
>
> ### Writing
> * We deeply apologize for our unclear writing. Thank you for your comment and we corrected them in the revision. We will make our best effort to revise the text in the final version.

---

> > ### Comment · Reviewer_BnBD · 2022-08-09
> > **Thank you**
> >
> > Thank you for your response!

---

### Official Review · Reviewer_WWGT · 2022-07-10

**Rating:** 6
**Confidence:** 2
**Soundness:** 3 good
**Presentation:** 3 good
**Contribution:** 3 good

**Summary:**

This paper proposes a novel 2D-3D Weakly Supervised Semantic Segmentation framework that jointly targets both domains of image and point cloud. To bridge the gap between 2D and 3D domains, the 3D outputs are projected onto the image plane. And they build the consistency of the correlation matrix of
2D features and the projected 3D features. Then they build the consistency of the 2D CAM and the projected 3D CAM.

**Questions:**

In consideration of the reproducibility of this work, the authors are suggested to provide their source code as supplementary.

In Figure 2, there is a projection function from the point cloud to the image. Is the image built from a point cloud? If not, please remove this projection function.

Why 3D-to-2D loss only affects the image branch, while the 2D-to-3D loss affects the point cloud branch? Can they conduct in two-ways? More comprehensive and in-depth studies are needed to justify this design.

**Ethics Review Area:**

["I don’t know"]

**Limitations:**

The proposed method requires the perspective projection matrix as input during training, which might be difficult to obtain in a real-world applications.

Only ResNet38 and Point Net++ are employed in this paper. I suggest authors add more backbone to justify the generalization ability of the proposed framework.


**Strengths And Weaknesses:**

(+) The idea is interesting and easy to follow.
(+) The ablation studies are comprehensive.
(+) SOTA performance on the ScanNetV2 and S3DIS datasets.
(-) Compared with the baseline, the proposed method requires the perspective projection matrix and images as input during training. This might cause an unfair comparison in the experiment.
(-) The Performance comparison with other state-of-the-art 2D WSSS methods e.g.[R1-R5] is missing.

[R1] Xu, Lian, et al. "Leveraging auxiliary tasks with affinity learning for weakly supervised semantic segmentation." Proceedings of the IEEE/CVF International Conference on Computer Vision. 2021.

[R2] Yao, Yazhou, et al. "Non-salient region object mining for weakly supervised semantic segmentation." Proceedings of the IEEE/CVF Conference on Computer Vision and Pattern Recognition. 2021.

[R3] Wu, Tong, et al. "Embedded discriminative attention mechanism for weakly supervised semantic segmentation." Proceedings of the IEEE/CVF Conference on Computer Vision and Pattern Recognition. 2021.

[R4] Lee, Jungbeom, et al. "Reducing information bottleneck for weakly

[R5] Zhou, Tianfei, et al. "Regional semantic contrast and aggregation for weakly supervised semantic segmentation." Proceedings of the IEEE/CVF Conference on Computer Vision and Pattern Recognition. 2022.

---

> ### Author Response · Authors · 2022-08-02
> **Response to Reviewer WWGT (1/2)**
>
> Thank you for your thoughtful and constructive comments. We revised our paper and supplementary materials and uploaded them to OpenReview. Below we address the comments in the review point by point.
>
> ### Requiring projection matrices and images for the training
> * Since the proposed method requires additional information (perspective projection matrix and images) during training, we deeply agree with you that it would be not fair to directly compare it with the other methods in terms of performance.
>
> * Please understand that our method paper proposes a practical task formulation. The existing (indoor) point cloud datasets such as ScanNetV2 or S3DIS have usually reconstructed the point cloud from a sequence of RGBD frames. Thus, such datasets usually simultaneously provide 3D/2D/projection matrix data, which is suitable for training our joint learning framework.
>
> * We want to emphasize that **this paper is the first attempt that targets multi-dimensional WSSS**, to the best of our knowledge. We not only formulate this novel and practical task but also propose a joint learning framework as a strong baseline. We believe that the field of 2D and 3D WSSS (that had been individually studied) can be potentially helpful for each other, and we would be glad if our work could be a bridge for it.
>
> ### Performance comparison with other SOTA 2D WSSS methods
> * As far as we know, the proposed framework is the first attempt to jointly learn the multi-dimensional WSSS. Therefore, in this paper, we placed more emphasis on outperforming the uni-dimensional (2D-only or 3D-only) baseline, rather than competing with the other SoTAs performing a single task. We believe that our framework can serve as a strong baseline for future works on multi-dimensional WSSS. Nevertheless, we thoroughly agree that the comparison with SoTA 2D WSSS methods is meaningful.
>
> * In fact, **there are critical technical issues with directly applying the existing SoTA 2D WSSS methods to the ScanNetV2 image dataset**. Most of the 2D WSSS studies have been conducted on PASCAL VOC 2012 or MS-COCO 2014 datasets, which are far different from the ScanNetV2 dataset. The images of PASCAL and COCO datasets include several foreground object classes, and the non-object regions are defined as a background. On the other hand, ScanNetV2 images are fully segmented by several indoor classes including background-like classes (e.g. wall or floor).
>
> * The difference might seem trivial; however, because of this, it is difficult to directly apply the conventional 2D WSSS methods on ScanNetV2 dataset. SoTA 2D WSSS methods usually employ pre-trained saliency modules for acquiring objectness cues to separate the foreground and background regions. However, in the case of ScanNetV2, the saliency module tends to regard background-like classes (e.g. wall or floor) as non-salient objects, and thereby suppresses their activation.
>
> * To avoid the issue, we selected two recent 2D WSSS methods [R6, R7] that do not use the pre-trained saliency module and are easy to access the code. R6 and R7 achieve 46.2% and 40.5% mIoU performance in ScanNetV2 train split, respectively. Although the 2D-only WSSS methods achieve improvements over the baseline (vanilla 2D WSSS, 38.0%), **our framework (54.4%) still outperforms the methods** by great margins.
>
> * We are aware of this comparison is not completely fair (since the proposed method utilizes 3D/pose data during training); however, the significant improvements in performance show that our framework could make a meaningful contribution to the field of WSSS.
>
> * Please generously understand that the rebuttal period is not enough to conduct an extensive comparison, and we made our best effort to address the comment. We will endeavor to add comparisons with more methods in the final revision.
>
> * References
> > *[R6] Kweon, Hyeokjun, et al. "Unlocking the potential of ordinary classifier: Class-specific adversarial erasing framework for weakly supervised semantic segmentation." Proceedings of the IEEE/CVF International Conference on Computer Vision. 2021.*
>
>     > *[R7] Chen, Zhaozheng, et al. "Class Re-Activation Maps for Weakly-Supervised Semantic Segmentation." Proceedings of the IEEE/CVF Conference on Computer Vision and Pattern Recognition. 2022.*
>
>
> ### Code for reproduction
> * We sincerely promise that we will provide our source code upon acceptance.

---

> ### Author Response · Authors · 2022-08-02
> **Response to Reviewer WWGT (2/2)**
>
> ### Is the image built from a point cloud?
> * We apologize for the misleading figure. The image is captured by a real camera from the real scene. The projection function from the point cloud to the image is the perspective matrix of the real camera.
>
> * Actually, we intended to deliver that the projection matrix for a 3D feature and 3D CAM is the same as that of the real camera (that captured the real image). We revised both the figure and its caption to clarify the points we attempted to make.
>
> ### Direction of the joint losses
> * We designed the 2D-to-3D loss to leverage the 2D CAM as self-supervision for improving the semantic perception of the 3D CAM. Likewise, the 3D-to-2D loss is devised to transfer the geometrical structure of the 3D scene to the 2D domain.
>
> * Following your constructive comment, we conducted an additional experiment. In this experiment, we set both 2D-to-3D and 3D-to-2D losses in a bidirectional way. We observe that this two-way design non-negligibly decrease the performance in mIoU (2D: from 54.4% to 43.2%, 3D: from 59.1% to 51.5%) on ScanNetV2 train split.
>
> * We think the reason is that the 2D and 3D networks have different strengths at initial, as we mentioned in the main paper. Our asymmetrical design for the loss function seems more effective and feasible to transfer the strengths of one domain to the other domain. With the aforementioned experiment and results, we will make our best effort to revise our paper in the final version.
>
> ### Backbones
> * We want to emphasize that the ResNet38 and PointNet++ are frequently employed backbones in the field of 2D and 3D WSSS, respectively. Likewise, for a fair comparison with the existing works, we employ ResNet38 and PointNet++ as our backbones.
>
> * Since the proposed framework does not have any explicit architectural constraints, we agree that the results with multiple different backbones would strengthen our paper. However, please generously understand that the rebuttal period is not enough to conduct the experiment. Thank you again for your constructive suggestions and we will endeavor to add the results with different backbones in the final version.

---

### Official Review · Reviewer_u5nu · 2022-07-11

**Rating:** 5
**Confidence:** 5
**Soundness:** 3 good
**Presentation:** 3 good
**Contribution:** 3 good

**Summary:**

This paper proposes a method to do weakly supervised semantic segmentation with 2D-3D information. Existing methods have studied the use of class activation maps (CAMs) on 2D images and 3D point clouds for semantic segmentation independently. This paper proposes to leverage 2D and 3D CAMs jointly to learn a network for better 3D semantic segmentation under weak supervision. A joint use of 2D and 3D CAMs can alleviate the sharpness issue of 2D CAMs at object boundaries with the geometric information from 3D point clouds, and similarity, can mitigate the class imbalance and data sparseness problems in 3D CAMs with 2D CAMs. To this end, a 3D-to-2D loss and a 2D-to-3D loss is proposed to enforce the consistency between the similarity matrix of the 2D and projected 3D features, and 2D pseudo label  and projected 3D CAM, respectively.

**Questions:**

1. Why is it named 2D-to-3D loss in Equation (7)? Isn't the 3D activation map projected into 2D for comparison with the 2D pseudo-labels obtained from the 2D activation map, and thus should be also a 3D-to-2D loss?
2. How much does the joint 2D-3D learning help weakly supervised 2D image segmentation? Intuitively from the explanations in the paper, 2D segmentation results should also improve, especially in the aspect of clear boundaries of the segmentations should be seen. This would be a clear proof that 3D geometry is useful for 2D segmentation, and joint learning is effective in both directions.
3. How is the experimental results conducted only on ScanNet-V2 convincing that the proposed method can work well on other datasets? Otherwise, it is important to show results on at least another dataset. Although the proposed method is convincing, the lack of experimental results on other dataset(s) is a major weakness of the paper.

**Limitations:**

Yes, the limitation is discussed, but the negative societal impact is not discussed in the paper.

**Strengths And Weaknesses:**

Strengths:
+ This paper is well-written, the core ideas are easy to follow.
+ The proposed framework of leveraging the information from 2D images to improve 3D semantic segmentation under weak supervision is interesting. It is really good to see that the paper is written in a way to highlight how the 3D geometrical information from point clouds and the rich and dense 2D image information can compliment each other in the training process.
+ The proposed 3D-to-2D loss on the class correlation matrix of the 2D image and the projected 3D class correlation matrix is technically sound. Similarly, the 2D-to-3D loss is a good idea to enforce the consistency between the 2D and 3D activation maps.
+ The image classifier pretrained on imageNet can also be seen as the transfer of 2D knowledge to 3D point clouds. Specifically, there is a lack of large-scale dataset for pretraining for 3D point clouds, and thus the use of 2D information from the large scale imageNet can effectively help 3D semantic segmentation.

Weaknesses:
- While it's understandable that the final goal is to improve 3D point cloud semantic segmentation, the proposed framework of joint 2D-3D learning should actually also help improve the results of 2D semantic segmentation. However, there is no experiments done to show that the 2D semantic segmentation actually benefited from the joint learning.
- The experiment results are only done on one dataset, i.e. the ScanNetV2 dataset. There should be other dataset with paired 2D image and 3D point cloud available for training and testing the proposed model. It also remains unclear how generalizable is the proposed method, e.g. how well it works on other dataset when it's trained on ScanNetV2? The incompleteness of the experiments somehow weakened the paper.
- The proposed method requires 2D image-to-3D point cloud pairs. It would be better if the method can also leverage unpaired images and 3D point clouds for training. However, this limitation seems to be difficult to alleviate as the method dependents heavily on the projection. of 3D to 2D.

---

> ### Author Response · Authors · 2022-08-02
> **Response to Reviewer u5nu (1/2)**
>
> Thank you for your thoughtful and constructive comments. We revised our paper and supplementary materials and uploaded them to OpenReview. Especially, we clarified that our framework achieves significant improvements over baselines in both 2D and 3D domains. Below we address the comments in the review point by point.
>
> ### Does 2D-3D joint learning help the 2D domain?
>
> * This point might not be clear in our submission, and we would like to apologize for the misleading. However, we want to politely clarify that our joint framework can perform 2D and 3D semantic segmentation separately, and the main paper had already included several experimental results showing that our joint 2D-3D learning actually helps weakly-supervised 2D image segmentation.
>
> * **Quantitatively**: the first row of Table 2 shows the performance of 2D image segmentation when we use classification loss only (our baseline for the 2D domain). Compared to the baseline, **the second row (additionally using 3D-to-2D joint loss) achieves much higher performance on both train/val sets in the 2D domain**. Also, the following rows strongly support that the proposed framework greatly helps WSSS in the 2D domain, as well as the 3D domain.
>
> * **Qualitatively**: Figure 3 compares the 2D CAM acquired from the different settings. (b1) uses mere classification loss only, which is our baseline. On the other hand, (b2) and (b3) employ 3D-to-2D loss and bidirectional joint loss, respectively. Compared to the baseline, **we can observe that our framework achieves clearer boundaries of the segmentation**. Furthermore, as shown in Figure 4, the proposed joint loss provides much better semantic segmentation results than our 2D baseline, in terms of segmentation boundaries. We provide more results in the supplementary material (Figures 2, 4, 5).
>
> * To sum up, both the quantitative and qualitative results strongly support that the proposed joint learning is effective not only in the 3D but also in the 2D domain. To provide clearer focus and comprehension, we revised the introduction.
>
> ### Generalization
>
> * Please note that the existing (indoor) point cloud datasets such as ScanNetV2 and S3DIS have usually obtained the point cloud from a sequence of RGBD frames. Since they use fixed-view sensors or BundleFusion during 3D reconstruction, such datasets usually provide 3D point clouds, 2D images, and the poses of the 2D images, which is suitable for training our joint framework. Nevertheless, we fully understand your concern about generalization, which is an important and practical issue in WSSS.
>
> * To address it, we experimentally verified the generalization capability of the proposed method using the **S3DIS dataset**, as in WyPR [22]. In this experiment, we first train our model on ScanNetV2. Then we test it using S3DIS by feeding the 3D point clouds only for 3D semantic segmentation and 2D images only for 2D semantic segmentation.
>
> * We provide the result in **Tables 3 and 4** of the Supplementary Materials. We can observe that the proposed framework achieves reasonable performance on S3DIS, outperforming our baseline by a large gap. The results strongly support that **our joint learning method has a degree of generalization capacity**.

---

> ### Author Response · Authors · 2022-08-02
> **Response to Reviewer u5nu (2/2)**
>
> ### Training in an unpaired setting
>
> * Surprisingly, we also initially considered training our framework in an unpaired setting. We are happy to see that reviewer suggests a research direction similar to our philosophy.
>
> * As you pointed out, the proposed 2D-3D joint loss cannot be defined without the projection matrix. **To overcome this limitation, we attempted to render images from a point cloud with randomly sampled camera parameters.** Here, the rendered images could be regarded as virtual pairs of the point cloud, where the projection matrix is known. Then, we trained our framework with a combined set of the real images and the rendered images. The 2D-3D joint loss is only applied to the rendered images.
>
> * The main limitation of this strategy was **a domain gap between the real images and the rendered images**. The rendered images include lots of holes and empty pixels due to the sparse nature of the point cloud. Although we observe sort of improvements in 3D performance, it was not that impressive due to the low reliability of 2D CAMs. We believe that incorporating domain adaptation to the aforementioned approach for relieving the domain would be really interesting future work.
>
> * To sum up, we humbly admit that the paired setting (in the training phase) is one of the limitations of our method. We agree with the necessity of incorporating the unpaired setting and would research more about it in the near future.
>
> ### Name of the loss function in Equation (7)
>
> * In Equation (7), the 3D activation map is projected into 2D and compared with the 2D pseudo-labels. Therefore, from some perspectives, it is understandable that the name of this loss should be 3D-to-2D loss.
>
> * However, since **the purpose of this loss is to transfer the semantic capability from the 2D domain to the 3D domain**, we believe that 2D-to-3D loss would be a more appropriate and intuitive name. (Similarly, the loss in Equation (11) is defined as 3D-to-2D loss since it aims to transfer segmentation capability from 3D to 2D.) Nevertheless, we agree that our naming (2D-to-3D loss) might lead readers to misunderstand the concept. We revised the text to clarify the points we attempted to make.

---

### Official Review · Reviewer_JTDc · 2022-07-11

**Rating:** 6
**Confidence:** 3
**Soundness:** 3 good
**Presentation:** 3 good
**Contribution:** 3 good

**Summary:**

This paper targets at weakly supervised semantic segmentation for both 2D and 3D domain. The intuition is that the weakly supervisedly features in these two domains are complementary and can benefit each other. Based on this intuition, the paper proposes two constraints, transferring the knowledge from 2D domain to 3D domain and vice versa. Specifically, the model uses 2D sudo segmentation label as supervision for 3D segmentation learning, while encouraging the 2D feature correlation map to match its 3D counterpart. During inference, the model does not need pair 2D-3D data, but instead carry out semantic segmentation on each domain separately.

**Questions:**

**Method.**

My foremost concern is that the proposed method needs paired 2D and 3D data, which is hard to obtain and such datasets usually already include ground truth 2D/3D semantic segmentation maps, e.g., ScanNet. Thus it is important and practical for the model to generalize to in-the-wild images or scenes. The model in the paper is learned on ScanNet, then the question is would there be a large domain gap if we apply the model to in-the-wild 2D images or 3D scenes?

**Experiment details.**
- In section 4.1, it seems the 2D &rarr; 3D and 3D &rarr; 2D consistency constraints are included from the beginning. I am curious how the model acts in the beginning, when 3D features are not good while the 2D part is pre-trained on ImageNet, will the 3D correlation maps be misleading in the beginning?
- How to decide the occluded object in line 137?

**Paper writing.**
- I would suggest briefly introduce the Class Activation Maps in the Introduction, which would be friendly for readers without a background. For now the paper mentioned the concept throughout the Introduction but doesn't properly explain it until line 111.
- Minor spelling erros:
  - line 3: has &rarr; have
  - line 42: can &rarr; to

**Limitations:**

The authors didn't discuss limitations or societal impact.

As stated above, I would suggest the authors to discuss the generalization capacity of the proposed method on images or scenes outside the chosen dataset.

**Strengths And Weaknesses:**

**Strengths.**
- This paper is well written and easy to follow.
- The proposed method is effective and improves existing methods by a large margin.

**Weaknesses.**
- The proposed method needs pair 2D-3D data for training, which is tricky to get. The paper doesn't demonstrate if the learned model is practical and can generalize to images or scenes outside the chosen dataset.

---

> ### Author Response · Authors · 2022-08-02
> **Response to Reviewer JTDc**
>
> Thank you for your thoughtful and constructive comments. We revised our paper and supplementary materials and uploaded them to OpenReview. Below we address the comments in the review point by point.
>
>
> ### Generalization
>
> * Please note that the existing (indoor) point cloud datasets such as ScanNetV2 and S3DIS have usually obtained the point cloud from a sequence of RGBD frames. Since they use fixed-view sensors or BundleFusion during 3D reconstruction, such datasets usually provide 3D point clouds, 2D images, and the poses of the 2D images, which is suitable for training our joint framework. Nevertheless, we fully understand your concern about generalization, which is an important and practical issue in WSSS.
>
> * To address it, we experimentally verified the generalization capability of the proposed method using the **S3DIS dataset**, as in WyPR [22]. In this experiment, we first train our model on ScanNetV2. Then we test it using S3DIS by feeding the 3D point clouds only for 3D semantic segmentation and 2D images only for 2D semantic segmentation.
>
> * We provide the result in **Tables 3 and 4** of the Supplementary Materials. We can observe that the proposed framework achieves reasonable performance on S3DIS, outperforming our baseline by a large gap. The results strongly support that **our joint learning method has a degree of generalization capacity**.
>
> ### Training procedure
>
> * First of all, we sincerely apologize for the unclear explanation of the training procedure. We revised the paper to include the following details.
>
> * In the first phase, we individually train both 2D and 3D classifiers with the classification loss of each domain. After that, we jointly train them using the proposed 2D-to-3D and 3D-to-2D losses, in addition to the classification loss.
>
> * Here, in the second phase, please note that we first train our framework without the 3D-to-2D loss for the first few epochs.
> Although we pre-trained the 3D network in the first phase and thereby it can extract meaningful 3D features for segmentation to some degree, it is true that the 2D network shows much better semantic segmentation capability in the first phase (refer to the 2D and 3D baselines in Table 4 of the main paper). Since this imbalance might lead to unstable joint training, we decided to strengthen the 3D network with 2D-to-3D loss in the early epochs of the second phase.
>
>
> ### Occlusion filtering
>
> * At the time of submission, due to the limited space, we put the explanation about the filtering process in the Supplementary Material. Nevertheless, we humbly admit to the reviewers' comment that the mention in the main paper (line 137) is too vague. In the revision, we elaborated on the part regarding the filtering process, with the following details.
>
> * For filtering the occlusion, we exploit the depth map corresponding to the image provided by the ScanNetV2 dataset as a reference. We could also render a depth map from the mesh (generated from the point cloud).
> Then, the points having the same depth as the reference depth are preserved during projection, while the other points are filtered.
>
>
> ### Writing
> * Thank you for the constructive suggestion. We agree that more explanation regarding the concept of CAM would be helpful for readers. We added it in the introduction in revision (line 27-28). Also, we apologize for the spelling errors and revised them.

---

### Meta-Review · Area_Chair_fQvi · 2022-08-22

**Recommendation:** Accept
**Confidence:** Certain

**Metareview:**

The paper proposes two constraints, transferring the knowledge from 2D domain to 3D domain and vice versa, to do weakly supervised semantic segmentation in 2D and 3D.  During inference the model doesn't require paired data.

The paper is clearly written, and explains how the 3D geometrical information complements the redundant dense 2D information.  The reviewers mention limited experiments.  However, the ablations are thorough, and show significant improvement over SotA performance.

The authors addressed the reviewers' comments during the rebuttal.  The reviewers agree that the benefits of the proposal outweigh its flaws.

I recommend the paper for publication.

**Award:**

No

---

### Decision · Program_Chairs · 2022-09-14

Accept